# Is Large-scale Pretraining the Secret to Good Domain Generalization?

**Piotr Teterwak**[1], **Kuniaki Saito**[1], **Theodoros Tsiligkaridis**[2], **Bryan A. Plummer**[1*], **Kate Saenko**[1*]

[1]Boston University      [2]MIT Lincoln Laboratory

{piotrt,kesaito,bplum,saenko}@bu.edu     tsili@ll.mit.edu

## Abstract

Multi-Source Domain Generalization (DG) is the task of training on multiple source domains and achieving high classification performance on unseen target domains. Recent methods combine robust features from web-scale pretrained backbones with new features learned from source data, and this has dramatically improved benchmark results. However, it remains unclear if DG finetuning methods are becoming better over time, or if improved benchmark performance is simply an artifact of stronger pre-training. Prior studies have shown that perceptual similarity to pre-training data correlates with zero-shot performance, but we find the effect limited in the DG setting. Instead, we posit that having perceptually similar data in pretraining is not enough; and that it is how well these data were learned that determines performance. This leads us to introduce the Alignment Hypothesis, which states that the final DG performance will be high if and only if alignment of image and class label text embeddings is high. Our experiments confirm the Alignment Hypothesis is true, and we use it as an analysis tool of existing DG methods evaluated on DomainBed datasets by splitting evaluation data into In-pretraining (IP) and Out-of-pretraining (OOP). We show that all evaluated DG methods struggle on DomainBed-OOP, while recent methods excel on DomainBed-IP. Put together, our findings highlight the need for DG methods which can generalize beyond pretraining alignment. We release DomainBed-OOP at https://huggingface.co/datasets/PTeterwak/DomainBed_OOP.

## 1 Introduction

Domain Generalization (DG) addresses the challenge of enabling AI models to generalize from known domains to unseen ones, a critical task given the inevitable distribution shifts between training and real-world deployment (Saenko et al., 2010). DG pipelines typically consist of three stages: pretraining a model on a large, general dataset; finetuning the model with one or more source domains; and finally evaluating the model on target domains that are distinct from source domains. DG methods increasingly rely on huge-scale foundation models for initialization (*e.g.*, (Shu et al., 2023; Cha et al., 2022; Addepalli et al., 2024)). Simultaneously, finetuning has increasingly incorporated regularization to prevent catastrophic forgetting. As a result, it remains unclear whether DG methods are genuinely improving or if benchmark performance gains are simply due to stronger pre-training, possibly with the target domains within the hundred million-scale pre-training data (Mayilvahanan et al., 2025), combined with regularization.

In this work, we examine the reliance of recent state-of-the-art CLIP-based DG methods on pretrained features. While prior studies have shown that perceptual similarity to pre-training data explains zero-shot performance—referred to as the Image Similarity Hypothesis (Mayilvahanan et al., 2024)—we find this relationship to be limited in the DG setting. Despite evidence of target domains being present in pre-training (Figure 4), we find that perceptual similarity alone does not fully explain accuracy in the DG context (Section 3). We argue that it is not just the presence of similar data in pre-training that matters, but also how well this data was learned. To this end, we introduce the Alignment Hypothesis, which states that pre-trained alignment between image and class

---

*Indicates equal contribution as senior author

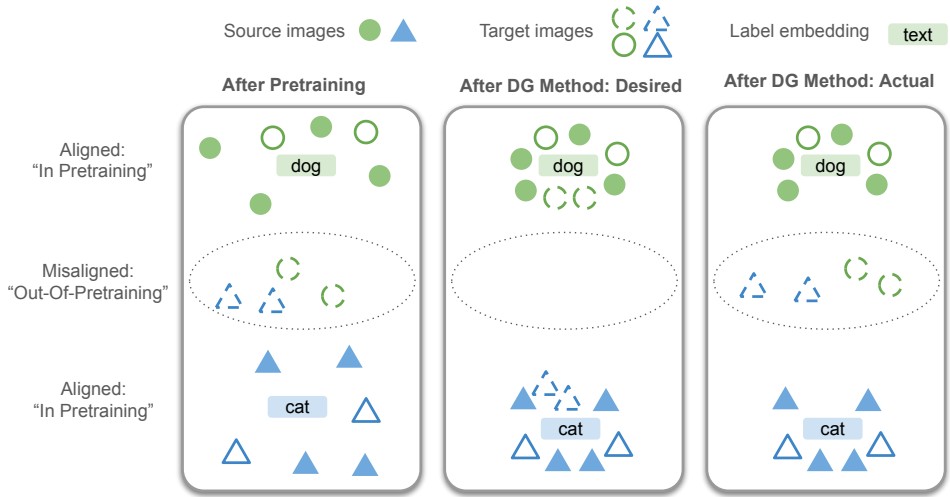

Figure 1: **An overview of desired and actual behaviour of DG methods. 1)** DG methods are initialized with foundation models like CLIP. Pre-trained embeddings are relatively well aligned with ground truth labels on both source and target data for most samples (In-Pretraining, IP), but some samples are not well aligned (Out-of-pretraining, OOP). **2)** An ideal DG method would strengthen alignment for both OOP and IP data with ground truth labels. **3)** Our analysis shows that DG methods only result in strong alignment for IP data, leaving OOP data misaligned (Figure 2).

embeddings is still predictive of DG performance even after source finetuning. We note that we do not make assumptions of how or why alignment arose. In the Alignment Hypothesis, pre-training alignmnent is used as a measure for how well a sample is learned. We find that performance for low alignment samples can be almost 0, while performance for high alignment samples is close to perfect. These results confirm the Alignment Hypothesis. As illustrated in Figure 1, these findings suggest that current DG methods largely fail to learn new, general features from the source data when the pretraining does not already provide a strong alignment.

The confirmation of the Alignment Hypothesis gives us a tool to separate aligned and well learned in-pretraining (IP) data from misaligned and poorly learned out-of-pretraining (OOP) data for a particular backbone, and we do so for five DG datasets with OpenCLIP-ViT/B-16. We call the resulting splits DomainBed-IP and DomainBed-OOP. Evaluating on DomainBed-IP/OOP offers a view of where current DG methods fail and where they succeed. We focus on CLIP-based methods, as they are used in state-of-the-art DG methods (Addepalli et al., 2024; Cho et al., 2023; Shu et al., 2023; Mao et al., 2024); we believe extensions to pure vision models such as DINOv2 (Oquab et al., 2023) represent interesting future work. We find that all methods, including those considered state-of-the-art, perform poorly on OOP data, *i.e.*, data that the pretrained backbone hadn't already aligned well. Furthermore, recent state-of-the-art methods do not outperform older methods on OOP data; CLIPood (Shu et al., 2023) slightly under performs a combination of older methods (MIRO (Cha et al., 2022) + MPA (Arpit et al., 2022)) on DomainBed-OOP. At the same time, existing DG methods show exceptional performance on DomainBed-IP, sometimes even outperforming an oracle model *trained on the target domain*. These results suggest that future research should aim to enhance DG methods on low-alignment data. In summary, we make the following contributions:

- **Introduce the Alignment Hypothesis**: We demonstrate that pre-training alignment between image and class text embeddings is a stronger predictor of Domain Generalization (DG) success than the previously proposed Image Similarity Hypothesis (Mayilvahanan et al., 2024). Based on this, we define In-Pretraining (IP) as data well-aligned with pretrained embeddings, and Out-of-Pretraining (OOP) as data with weaker alignment.
- **Propose a new IP/OOP evaluation framework**: We demonstrate that splitting target data by its alignment with the pre-trained backbone can effectively test Out-of-Pretraining (OOP) generalization. We release IP/OOP splits for DomainBed datasets.
- **Expose strengths and limitations of state-of-the-art DG methods**: Using DomainBed-IP/OOP we find that leading DG methods perform well on data well-aligned by pre-training

but struggle on misaligned samples, emphasizing the need for methods that move beyond reliance on pre-training.

## 2   RELATED WORK

**Multi-Source Domain Generalization:** Domain Generalization aims to mitigate the impacts of domain shifts between source (training) and target (deployment) domains. These can include sub-population shifts where all testing subpopulations are present in the training data but in different proportions (Dehdashtian et al., 2024), or it could be the case we consider in this work where the testing subpopulation is not at all present in the training subpopulation. Although we focus on the multi-source domain generalization task, where domains between train and test are completely disjoint, our analysis can be extended to other types of generalization. One standard approach is **domain-invariant feature learning**, which leverages domain labels to learn domain-invariant features. CORAL (Sun & Saenko, 2016) aligns second-order statistics, while DANN (Ganin et al., 2016) and ADDA (Tzeng et al., 2017) uses an adversarial loss. Gulrajani & Lopez-Paz (2020) show that ERM, which does not align features between domains, can outperform most prior work while being easier to tune. Another common approach is **domain-aware data augmentation** to expand the training domain to become closer to or even overlap the target domain. Inter-domain mixup (Yan et al., 2020) blends images from different domains. Similarly, style transfer can diversify training images (Zhong et al., 2022). **Deep ensembles** are effective for domain generalization (Arpit et al., 2022). Since they are computationally inefficient for inference, many recent works average model weights from either multiple finetuning runs or from a single training trajectory (Cha et al., 2021; Arpit et al., 2022; Rame et al., 2022; Jain et al., 2023; Li et al., 2023; Shu et al., 2023). More recently, several methods perform **regularized finetuning** towards the initialization of a pretrained model. This works under the assumption that pretrained features are useful for target data, and should not be unlearned. The general idea can be applied to weight space (L2SP (Xuhong et al., 2018)), feature space (MIRO (Cha et al., 2022)), or output space (*e.g.*, CAR-FT (Mao et al., 2024), CLIPood (Shu et al., 2023)).

**Large-Scale Pretraining for DG:** Recent DG literature (Cho et al., 2023; Cha et al., 2022; Addepalli et al., 2024; Mao et al., 2024; Arpit et al., 2022) leverages large-scale pretrained initializations stronger than ImageNet (Russakovsky et al., 2015), and CLIP (Radford et al., 2021) is the most common choice. CLIP leverages a cross-domain contrastive loss to align images and captions. Due to the large scale of training data (typically at least 400 million samples) and the free-form nature of the text, CLIP enables effective zero-shot classification and learns features that generalize very well. Other choices for very strong pretraining include SWAG (Singh et al., 2022) and DinoV2 (Oquab et al., 2023). SWAG uses supervision from Instagram hashtags, while DinoV2 is trained without text supervision and instead relies on augmentation-based alignment. While our analysis focuses on image-text models like CLIP due to its popularity, the concept of alignment can extend to other types of pretraining models. We leave the exploration of this extension to future work.

**Impact of Data on Model Performance:** Several recent studies have explored the influence of pre-training data on model performance. Mayilvahanan et al. (2024) investigated how the presence of perceptually similar images in CLIP (Radford et al., 2021) pretraining affects performance, introducing the Similarity Hypothesis, which posits that nearest neighbor similarity is strongly correlated with zero-shot accuracy. Mayilvahanan et al. (2025) show that domain contamination in the pretraining has substantial impact on DG performance. Udandarao et al. (2024) demonstrated that concept frequency in pretraining is correlated with zero-shot performance and introduced a dataset focusing on infrequent concepts. Fang et al. (2022) found that diversity in pretraining data is critical for improving performance on benchmarks such as ImageNetV2 (Recht et al., 2019), ImageNet-R (Hendrycks et al., 2021), ImageNet-Sketch (Wang et al., 2019), and ObjectNet (Barbu et al., 2019). However, these studies focus on the zero-shot setting, where models are evaluated without further training. In contrast, we examine the domain generalization setting, where pre-trained models are fine-tuned on source domains and tested on held-out target domains. Yu et al. (2024) recommend using self-supervised pre-training to avoid data label leakage. In contrast we study DG model behavior in the more realistic setting of CLIP-pretraining. Our findings suggest that comparing target images to pre-trained images, as proposed by Mayilvahanan et al. (2024), is less predictive of final DG performance than directly measuring the alignment between the image and its class embedding.

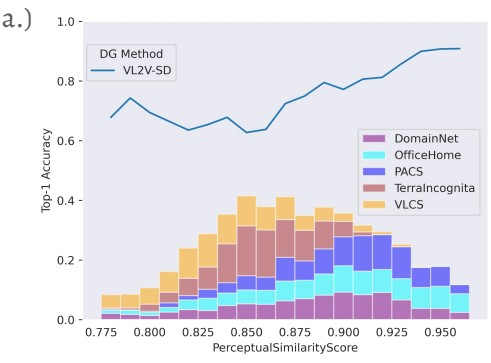 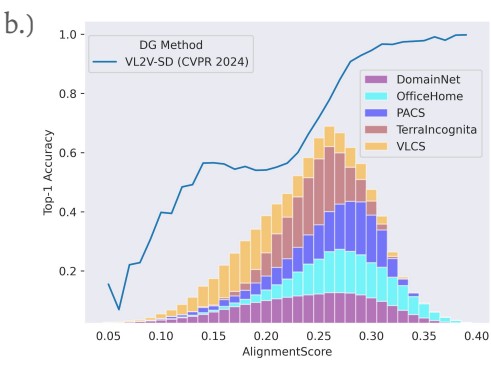

Figure 2: **Comparing the Predictive Power of the Alignment and Image Similarity Hypotheses for Domain Generalization (DG). a.) Image Similarity Hypothesis:** The cosine similarity between a test image and its closest match from the pre-training set (Perceptual Similarity Score) shows relatively weak predictive power for accuracy, implying that visual resemblance alone is not fully indicative of downstream performance. **b.) Alignment Hypothesis**: In contrast, the cosine similarity between image and ground truth text-label embedding after pre-training (Alignment Score) is highly predictive of model accuracy after fine-tuning on five DG datasets, with Alignment Score distributions shown in the colored histograms. This suggests that image-text pairs well-aligned during pre-training result in better performance on target tasks.

## 3   ANALYZING THE ROLE OF PRETRAINING IN DOMAIN GENERALIZATION

This work explores Multi-Source Domain Generalization for classification, where samples from multiple source domains (*e.g.*, sketches, product photos) and a held-out target domain (*e.g.*, wildlife camera images) are annotated with both domain and class labels. We construct a training dataset by aggregating all sample-label pairs from all training domains $d \in \{d_1, \ldots, d_n\}$, denoted as

$$D = \{(X^{d_1}, Y^{d_1}), \ldots, (X^{d_n}, Y^{d_n})\}.$$

We initialize a classifier $f$ with a contrastively pre-trained vision-language model (*e.g.*, CLIP) and finetune it on $D$. The scale of pre-training datasets is many orders of magnitude larger than that of source datasets. Most methods fully fine-tune $f$, though LP-FT (Kumar et al., 2021) fine-tunes the linear probe before the main network and Attention Tuning (Teterwak et al., 2023; Touvron et al., 2022) only tunes attention layers. The performance is then evaluated on a held-out testing domain $d_{\text{test}}$. The key assumption is that $d_{test}$ has a different distribution from the source domains.

We aim to analyze how reliant existing DG methods are on pre-training. A recent analysis of CLIP proposed the Image Similarity Hypothesis (Mayilvahanan et al., 2024), which supposes that high CLIP performance on a given test sample is a result of highly similar nearest-neighbor images in pre-training, and tested it on zero-shot classification tasks. They found a strong correlation between nearest-neighbor similarity and zero-shot classification performance, but did not analyze OOD performance after fine-tuning. Therefore, we apply an equivalent testing setup for the DG setting, where a pre-trained model is fine-tuned on a source distribution and tested on a different target distribution. We find only a limited influence of image similarity in Section 3.1. To better understand the role of pretraining in domain generalization, we introduce the Alignment Hypothesis, which we explore in detail in Section 3.2. We later use the Alignment Hypothesis to split DG datasets and analyze existing DG methods (Section 4).

### 3.1   TESTING THE IMAGE SIMILARITY HYPOTHESIS

The **Image Similarity Hypothesis** (Mayilvahanan et al., 2024) posits that test performance improves when there are perceptually similar images in the pre-training dataset. The PerceptualSimilarityScore measures perceptual similarity and is defined as the cosine similarity between a target image $I$ and its nearest neighbor $I_k$ in pre-training:

$$\text{PerceptualSimilarityScore}(I, I_k) = \frac{\langle f_I(I), f_I(I_k) \rangle}{\|f_I(I)\| \cdot \|f_I(I_k)\|} \qquad \text{(Eq. 1)}$$

---

**Algorithm 1** Evaluating the Image Similarity Hypothesis

---

**Require:** Target domain samples $D_{\text{target}}$, trained DG model $M$, pre-trained image encoder $f_I$,
 1: **for** each sample $I \in D_{\text{target}}$ **do**
 2:     Retrieve nearest neighbor of $I$ in LAION-400M using $f_I$ features, assign to $I_k$
 3:     Compute PerceptualSimilarityScore$(I, I_k)$ using Equation Eq. 1,
 4:     Record correctness of $M(I)$
 5: **end for**
 6: Bin samples based on PerceptualSimilarityScore
 7: Compute DG accuracy within each bin
 8: **return** Accuracy for each bin

---

where $\langle \cdot, \cdot \rangle$ denotes the dot product, and $\| \cdot \|$ denotes the Euclidean norm (magnitude). To evaluate the Image Similarity Hypothesis, we bin held-out target domain samples from five DomainBed datasets based on their PerceptualSimilarityScore and compute the accuracy of a Domain Generalization (DG) method independently for each bin. This procedure is detailed in Algorithm 1 and visualized in Figure 2. The PerceptualSimilarityScore is computed using approximate nearest neighbors over LAION-400M (Schuhmann et al., 2021) with the CLIP-retrieval library (Beaumont, 2022).

Figure 2 a.) shows the results of this analysis of the recent, high-performing *VL2V-SD* (Addepalli et al., 2024) on various DG datasets. While the Image Similarity Hypothesis is somewhat predictive of DG performance, its influence is not very strong. This suggests that perceptually similar pretraining data alone may not guarantee high DG performance; additional factors, such as how effectively the nearest neighbors were aligned with the target concept, may also be significant.

### 3.2 Introducing the Alignment Hypothesis

To find a stronger predictor of DG accuracy than perceptual similarity, we focus on how effectively pre-training captures the relationship between an image and its label. This leads us to propose the **Alignment Hypothesis**, which states that if an input image and its corresponding text label (*e.g.*, 'A photo of a {cls}') are well-aligned in the embedding space, final DG performance will be high. Crucially, alignment is measured before source fine-tuning while DG performance is measured after adaptation. This allows us to isolate the contribution of fine-tuning. Since models like CLIP optimize image-text pairs using a contrastive loss, cosine similarity between image and text embeddings is an alignment measure well coupled to their training objective. Therefore, we use it as our metric of pre-training generalization. More formally:

$$\text{AlignmentScore}(I, T) = \frac{\langle f_I(I), f_T(T) \rangle}{\| f_I(I) \| \cdot \| f_T(T) \|} \qquad \text{(Eq. 2)}$$

where $f_I(I)$ is the embedding of the image before finetuning on source, and $f_T(T)$ is the embedding of the text. Relative to directly using the contrastive loss value which also depends on negatives, PerceptualSimilarityScore has a scale which aligns across datasets (Appendix B.6).

We verify the Alignment Hypothesis similarly to the Image Similarity Hypothesis, by binning samples using the AlignmentScore and computing accuracy for each bin using VL2V-SD. We provide the same analysis for many more DG methods in the Appendix Figure 11. In Figure 2 b.), we can see that the Alignment Hypothesis explains DG performance after source finetuning, significantly more strongly than for the Image Similarity Hypothesis in Figure 2 a.) This finding suggests that source fine-tuning in DG, which aims to achieve high performance across all target samples, only succeeds on those with high initial alignment.

## 4 Re-thinking Domain Generalization Benchmarking using the Alignment Hypothesis

Knowing that the Alignment Hypothesis holds for contrastively trained image-text models (Section 3), we can now use it as a tool to probe the performance of DG methods across different levels of pre-training alignment. We apply this approach to five widely-used DomainBed (Gulrajani & Lopez-Paz, 2020) DG datasets: VLCS (Fang et al., 2013), PACS (Li et al., 2017), OfficeHome (Ganin et al.,

| AlignmentScore \\ Dataset | 0.0 - 0.1 | 0.1 - 0.2 | 0.2 - 0.3 | 0.3 - 0.4 | 0.4 - 0.5 |
|---|---|---|---|---|---|
| DomainNet | Bird  | Lipstick  | Table  | Fish  | Eraser  |
| TerraIncognita | N/A | Coyote  | Dog  | Bobcat  | N/A |
| PACS | Person  | Giraffe  | House  | Elephant  | N/A |
| VLCS | Chair  | Dog  | Person  | Car  | N/A |
| OfficeHome | Candles  | Monitor  | Bike  | Knives  | Clipboard  |

Figure 3: Representative DomainBed dataset samples and their labels at various AlignmentScore values. At very low AlignmentScores, most labels (red boxes) are incorrect. At very high AlignmentScores, text present in the image corresponds to the label.

2016), TerraIncognita (Beery et al., 2018), and DomainNet (Peng et al., 2019). This section discusses how we create new splits for existing DG datasets using AlignmentScore. We start by computing AlignmentScore for all samples in 5 DG datasets (Figure 3). Based on our observation that some samples are mislabeled, we perform dataset cleaning (Section 4.1). Then we find an AlignmentScore threshold to split DG datasets into well aligned In-Pretraining (IP) and poorly aligned Out-of-Pretraining (OOP) evaluation subsets (Section 4.2), which we later use for evaluating DG methods (Section 5). In order to connect the AlignmentScore with the DG method, we use the same backbone both for splitting the datasets into IP and OOP subsets and for training DG methods.

## 4.1 DATA EXPLORATION AND CLEANING

We start by visualizing the data of all the datasets at various AlignmentScore values. We show some representative samples in Figure 3. At very low scores, we find that a large fraction of labels are incorrect (red boxes in Figure 3). Thus, we divide the data into AlignmentScore intervals (*e.g.*, 0.00-0.05, 0.05-0.10, and so on, up to 0.2) and randomly sample 100 instances from each interval for every dataset. This allows us to systematically analyze the relationship between AlignmentScore and label accuracy across different score ranges. For each interval, we then count the fraction of mislabelled samples to better understand how low AlignmentScores are associated with labeling errors. We find that below an AlignmentScore of 0.15, label noise is unacceptably high, with all datasets suffering the most from mislabelling (Table 7 in Appendix). Therefore, we discard all samples with AlignmentScore less than 0.15 in DomainBed-IP/OOP. As shown in Table 8 (in the Appendix), we observe that the percentage of discarded samples due to mislabeling varies across datasets, with VLCS and DomainNet having the highest rates at 12.41% and 7.64%, respectively.

Furthermore, on the right side of Figure 3, we observe that at very high AlignmentScores (greater than 0.4), images often contain text directly related to the label. However, our goal is to evaluate visual recognition rather than text recognition (OCR), and CLIP is known to have strong OCR abilities (Fort, 2021), so we exclude all samples above AlignmentScore of 0.4 from DomainBed-IP/OOP. As shown in Table 8 (in the Appendix), only a small portion of data is removed due to OCR filtering (0.00-0.15% across datasets), but this issue may become more significant in future studies.

## 4.2 Data Splitting

After filtering, we focus on determining a threshold to split the dataset into In-Pretraining (IP) and Out-of-Pretraining (OOP) subsets. We select 0.21 as the threshold, based on the trends observed in Figure 2 b.), as this is the point where performance begins to improve significantly, indicating that existing methods become more effective. While this threshold represents a somewhat subjective choice informed by observed patterns, we provide AlignmentScores in the released data, allowing researchers the flexibility to experiment with their own thresholds.

Figure 18 in the Appendix shows how this split impacts the size and composition of each dataset. For example, clipart in DomainNet is predominantly categorized as IP, likely due to its frequent presence on the internet and, therefore, web-scraped pre-training data. In comparison, TerraIncognita-OOP is more balanced domains, but exhibits substantial class shift between IP and OOP splits (Figure 17 in the Appendix), meaning some classes are better aligned than others during pre-training.

We further investigate the distinctions between the IP and OOP subsets using VisDiff (Dunlap et al., 2024), an LLM-based system that identifies differences between sets of images. For each combination of dataset, domain, and class—except for DomainNet, where we subsample roughly 15% of the combinations due to computational constraints—we independently sample up to 30 images from both the IP and OOP subsets. VisDiff employs CLIP (Radford et al., 2021) to compute an AUC-ROC score for the natural language differences proposed by the LLM, and we retain only those differences with an AUC score of 0.7 or higher. Results are presented in Appendix Table 9.

Several clear patterns emerge from our analysis. Contextual or environmental elements often overshadow the primary object indicated by the text label. For example, in VLCS's SUN09 domain, the OOP subset for the car class frequently features images described as "historical architecture," while in Office Home the real-domain bed images are deemed OOP if they include "child-themed decor." These findings imply that object size may also play a role. In the car example, visible architecture implies that the car occupies only a small portion of the image, and in the bed example, the presence of child-themed decor indicates that the entire bedroom is visible rather than just the bed. To test this hypothesis, we computed the object size of the ground truth labeled objects using bounding boxes generated by the open-world object detector OWLv2 (Minderer et al., 2023). We found object size is correlated with the AlignmentScore (see Appendix Figure 10), with the sole exception being TerraIncognita. This indicates that even the presence of small wildlife can substantially increase the AlignmentScore, and that the background vegetation does not present a significantly conflicting signal that is sometimes present in other datasets. Overall, our findings suggest that pre-training which better represents scenes with multiple objects or concepts may improve benchmark performance.

## 5 Experiments

We evaluate Domain Generalization (DG) performance across both pretraining-aligned (IP) and pretraining-misaligned (OOP) data. We adhere to the DomainBed evaluation methodology, where one domain is chosen as the target, and the remaining act as source domains. To maintain a sufficient amount of training data, we train all DG methods on the original, unsplit datasets. We use hyper-parameter values recommended by the original implementation authors for each method.

After training, models are evaluated separately on IP and OOP subsets, as well as the original, unsplit test domain. This allows us to measure how well each method generalizes to both pretraining-aligned and pretraining-misaligned data. We follow the literature's standard practice of computing performance per target domain and averaging the results across all domains. This method ensures that any domain imbalances do not disproportionately influence the final performance metrics.

### 5.1 Algorithms

**Explicit Regularization towards Pretraining**: Several recent DG methods leverage explicit regularization towards the initialization. These methods generally operate either in weight space (by regularizing or freezing model parameters) or in feature space (by aligning internal feature representations with those of the pretrained model). *MIRO* (Cha et al., 2022) minimizes the Mutual Information between DG model intermediate features and CLIP intermediate features. *Attention Tuning* (Teterwak et al., 2023; Touvron et al., 2022) freezes all parameters except those in the

Multiheaded-Attention Layers. *VL2V-SD* (Addepalli et al., 2024) self-distills a linear combination of CLIP vision and text outputs into a model. *CLIPood* (Shu et al., 2023) regularizes both weights and output features. Weights are averaged between the pre-trained CLIP model and the fine-tuned model, and outputs are regularized using a loss that incorporates information from the pre-trained text encoder. *Linear Probe - Fine Tuning (LP-FT)* (Kumar et al., 2021) freezes the backbone, trains a linear probe, and then performs full finetuning. An untrained linear probe can cause finetuning to update the frozen backbone needlessly, potentially unlearning discriminative features. This biases the model towards pre-trained weights with smaller gradient updates.

**Domain Invariance**: A classic idea for Domain Generalization is Domain Invariance in the feature space, where the model learns only class-discriminative features shared among all training/source domains. *CORAL* (Sun & Saenko, 2016) matches the second moments of features across different domains and has been shown to be highly effective (Gulrajani & Lopez-Paz, 2020).

**Flat Optima**: Studies by Izmailov et al. (2018) and Cha et al. (2021) have shown that flat minima generalize better than sharp minima, as they make loss values less sensitive to perturbations in the loss surface, resulting in smaller increases in loss during domain shifts. *SWAD* (Cha et al., 2021) Averages model parameters during training, determining the interval over which to average using validation loss over source domain. *Model Parameter Averaging (MPA)* (Arpit et al., 2022) Starts averaging model parameters after a number of burn-in steps to find flat minima. *SAGM* (Wang et al., 2023) is an optimizer that explicitly optimizes for flat minima.

**Baseline and Oracle**: We also train a baseline and oracle model for lower-bound and upper-bound reference. The baseline model is an Empirical Risk Minimization (ERM) model that is finetuned on source domains and evaluated on target domains, and has been found to be effective for the DG task (Gulrajani & Lopez-Paz, 2020). The oracle model is trained on an 80% training split of all domains and evaluated on a 20% test split. The oracle model removes the OOD aspect of generalization and provides a reasonable upper bound for DG methods. Finally, we add a zero-shot LAION-400M (Schuhmann et al., 2021) pre-trained ViT-B/16 from OpenClip, which is the pre-trained model used in our analysis.

## 5.2 RESULTS

Table 1 reports results for the poorly pre-training aligned DomainBed-OOP, higly pre-training aligned DomainBed-IP, and standard DomainBed datasets. Underlined results highlight the best performance of any single method (excluding method combinations), while the bold numbers show the highest performance overall, excluding the upper bound and zeros-hot baseline. Overall, we find that AlignmentScore is highly predictive of performance. More detailed observations about the results follow, while a comparison to PerceptualSimilarityScore can be found in Appendix B.5.

**DG methods perform well on DomainBed-IP:** In most datasets within DomainBed-IP, domain generalization approaches achieve excellent performance. *On two out of five datasets ( OfficeHome and PACS), the best DG method even outperforms the oracle.* A notable exception is TerraIncognita, where CLIPood scores 30% below the oracle, highlighting this dataset as a challenging outlier. Interestingly, all three datasets where performance exceeds the oracle have an average IP AlignmentScore of 0.28, while the others have a lower average AlignmentScore of around 0.26. Therefore, the underperformance of TerraIncognita may be partially explained by its lower IP AlignmentScore, suggesting that alignment plays a significant role in DG performance, even within the IP case. Another interesting observation is that on the IP subset, the zero-shot model can achieve performances greater than the finetuned models (for PACS and VLCS). *This means that, for IP-data, DG finetuning sometimes causes more catastrophic forgetting than learning of new features* We show additional analysis of zero-shot models in Appendix B.9.

**DG Methods leave much to be desired on DomainBed-OOP, but are still stronger than ERM:** In DomainBed-OOP, we observe that even the top-performing DG methods struggle with low-alignment data. For example, CLIPood achieves 57.1% accuracy, which is a significant drop compared to its performance on DomainBed-IP (84.7%) and DomainBed-All (78.1%). Despite this, DG methods still outperform Empirical Risk Minimization (ERM) by up to 10% on DomainBed-OOP. Thus, DG methods are better equipped to handle domain shifts than ERM, possibly due to weak transfer of knowlegdge from pre-trained features. Nevertheless, there is still substantial room for improvement on low-alignment samples.

Table 1: Benchmarking DG methods on DB-IP/OOP. Domain generalization methods excel on high-alignment (IP) datasets—often even surpassing the oracle—while their performance noticeably drops on low-alignment (OOP) data, though still outperforming ERM.

| DomainBed-IP | DomainNet | OfficeHome | PACS | TI | VLCS | Average |
|---|---|---|---|---|---|---|
| OpenCLIP ZS | 74.9 | 88.9 | 98.5 | 36.8 | 95.9 | 79.0 |
| CORAL (Sun & Saenko, 2016) | 63.3 | 76.1 | 84.3 | 42.9 | 86.5 | 70.6 |
| SAGM (Wang et al., 2023) | 64.3 | 79.5 | 90.1 | 44.0 | 88.0 | 73.2 |
| ERM* (Gulrajani & Lopez-Paz, 2020) | 63.1 | 78.1 | 87.1 | 42.0 | 85.3 | 71.1 |
| LP-FT (Kumar et al., 2021) | 64.4 | 78.5 | 90.3 | 40.9 | 86.0 | 72.0 |
| SWAD (Cha et al., 2021) | 72.3 | 84.8 | 94.6 | 52.7 | 88.5 | 78.6 |
| MIRO (Cha et al., 2022) | 72.4 | 88.8 | 97.6 | 58.9 | 91.0 | 81.7 |
| VL2V-SD (Addepalli et al., 2024) | 78.1 | **91.4** | 98.0 | 48.1 | 92.4 | 81.6 |
| Attn Tune (Teterwak et al., 2023) | 69.2 | 84.8 | 96.4 | 53.0 | 88.7 | 78.4 |
| MPA (Arpit et al., 2022) | 73.6 | 85.1 | 95.4 | 54.4 | 90.7 | 79.8 |
| CLIPOOD (Shu et al., 2023) | **78.9** | 90.9 | 97.7 | **63.5** | **92.5** | **84.7** |
| MIRO + SWAD | 77.0 | 90.5 | 97.6 | 62.1 | 91.1 | 83.6 |
| MIRO + MPA | 78.2 | 90.7 | **98.1** | 62.6 | 91.0 | 84.1 |
| Upper Bound (Target Finetune) | 81.6 | 88.5 | 97.8 | 93.4 | 93.8 | 91.0 |

(a) Samples with high AlignmentScore values, indicating good pretraining alignment.

| DomainBed-OOP | DomainNet | OfficeHome | PACS | TI | VLCS | Average |
|---|---|---|---|---|---|---|
| OpenCLIP ZS | 26.3 | 48.1 | 81.4 | 4.5 | 80.2 | 48.1 |
| CORAL (Sun & Saenko, 2016) | 22.3 | 42.6 | 74.1 | 16.0 | 74.0 | 45.8 |
| SAGM (Wang et al., 2023) | 23.0 | 44.5 | 74.2 | 19.3 | 73.3 | 46.9 |
| ERM* (Gulrajani & Lopez-Paz, 2020) | 22.3 | 42.9 | 76.9 | 16.5 | 76.4 | 47.0 |
| LP-FT (Kumar et al., 2021) | 22.7 | 43.4 | 78.6 | 23.1 | 70.7 | 47.7 |
| SWAD (Cha et al., 2021) | 28.6 | 49.9 | 79.1 | 21.0 | 77.0 | 51.1 |
| MIRO (Cha et al., 2022) | 28.4 | 56.6 | 84.7 | 18.5 | 73.7 | 52.4 |
| VL2V-SD (Addepalli et al., 2024) | 31.8 | 56.6 | 85.0 | 15.9 | 79.1 | 53.7 |
| Attn Tune (Teterwak et al., 2023) | 26.8 | 51.4 | 84.2 | 20.3 | 76.1 | 51.8 |
| MPA (Arpit et al., 2022) | 29.6 | 51.0 | 82.7 | 22.2 | 79.5 | 53.0 |
| CLIPOOD (Shu et al., 2023) | **33.9** | **63.9** | 87.2 | 19.9 | **80.7** | 57.1 |
| MIRO + SWAD | 32.0 | 59.0 | 85.4 | 21.1 | 78.9 | 55.3 |
| MIRO + MPA | 33.1 | 60.0 | **87.8** | **24.9** | 80.3 | **57.2** |
| Upper Bound (Target Finetune) | 48.8 | 61.9 | 92.9 | 83.2 | 92.4 | 75.8 |

(b) Samples with lower AlignmentScore values, representing cases where pretraining alignment is weak.

| DomainBed-All | DomainNet | OfficeHome | PACS | TI | VLCS | Average |
|---|---|---|---|---|---|---|
| OpenCLIP ZS | 59.5 | 85.4 | 97.0 | 33.2 | 82.4 | 71.5 |
| CORAL (Sun & Saenko, 2016) | 50.6 | 73.2 | 83.2 | 39.6 | 78.5 | 65.0 |
| SAGM (Wang et al., 2019) | 51.5 | 76.4 | 87.5 | 41.0 | 80.4 | 67.3 |
| ERM* (Gulrajani & Lopez-Paz, 2020) | 50.5 | 75.0 | 85.2 | 39.0 | 77.9 | 65.5 |
| LP-FT (Kumar et al., 2021) | 51.3 | 75.5 | 88.4 | 38.5 | 78.0 | 66.3 |
| SWAD (Cha et al., 2021) | 57.9 | 81.8 | 92.4 | 49.0 | 80.1 | 72.2 |
| MIRO (Cha et al., 2022) | 57.5 | 85.8 | 96.4 | 54.3 | 81.1 | 75.0 |
| VL2V-SD (Addepalli et al., 2024) | 62.0 | **88.3** | 96.9 | 44.4 | 82.7 | 74.9 |
| Attn Tune (Teterwak et al., 2023) | 55.4 | 81.9 | 95.4 | 49.1 | 81.8 | 72.7 |
| MPA (Arpit et al., 2022) | 58.9 | 82.0 | 94.3 | 50.7 | 82.3 | 73.6 |
| CLIPOOD (Shu et al., 2023) | 63.6 | **88.3** | 96.8 | 58.5 | 83.4 | 78.1 |
| MIRO + SWAD | 61.4 | 87.6 | 96.6 | 57.4 | 82.0 | 77.0 |
| MIRO + MPA | 62.4 | 87.9 | **97.2** | 58.2 | 82.8 | 77.7 |
| Upper Bound (Target Finetune) | 70.4 | 86.2 | 97.2 | 92.4 | 87.9 | 86.8 |

(c) Performance of DG methods on unsplit DomainBed

**SOTA methods do not consistently outperform older methods on OOP data:** While CLIPood (Shu et al., 2023) clearly outperforms other methods on DomainBed-All, it performs comparably to older methods on DomainBed-OOP. For example, MIRO + MPA (Cha et al., 2022; Arpit et al., 2022) achieves 57.2% on DomainBed-OOP, which is nearly the same as CLIPood's 57.1%. This suggests that CLIPood's primary advantage comes from well-aligned samples in DomainBed-IP, where obtains 0.5% better performance than the next-best method.

**Model Parameter Averaging (MPA) boosts performance on OOP data**: MPA obtains a significant 6% gain over ERM on DomainBed-OOP. MPA is 0.5% better than MIRO on OOP data despite MPA being 2% worse than MIRO on IP data. When combined with MIRO, MPA delivers the best performance on DomainBed-OOP, slightly surpassing CLIPood. This suggests that MPA can complement other regularization-based methods like MIRO. On DomainBed-IP, MIRO + MPA is less than 1% away from CLIPood's performance, demonstrating versatility across both high- and low-alignment data. Interestingly, SWAD underperforms MPA on DomainBed-OOP by 2%, despite being conceptually similar. We attribute this to selecting the averaging interval on the source data, which introduces overfitting to source domains.

## 5.3 DISCUSSION

As an increasing number of works in the Domain Generalization sub-field leverage pre-trained CLIP models for Domain Generalization benchmarks, it is important to better characterize the impacts of pre-training on DG. We leave the reader with the following takeaways:

**Pre-training Alignment Predicts DG Performance:** Our study demonstrates that pre-training alignment, measured as the cosine similarity between image and text embeddings, is a robust predictor of DG performance. This holds true even after source fine-tuning, highlighting that the quality of alignment achieved during pre-training has a significant impact on the generalization capability.

**Current DG Methods Exploit Pre-training Rather Than Learning New Features:** Our findings reveal a large difference in the performance of DG methods between pretraining-aligned (IP) and pretraining-misaligned (OOP) data. While state-of-the-art methods achieve near-oracle performance on IP data, they struggle significantly on OOP data. This indicates that current methods primarily leverage on pre-trained features rather than learning new, generalizable features from source data. Consequently, their success is heavily tied to the quality of pre-training, rather than fine-tuning.

**Benchmarks Should Reflect Pre-training Reliance:** The reliance on pre-trained alignment calls for a reevaluation of DG benchmarks. Existing benchmarks often aggregate results across all target data, masking the limitations of DG methods on low-alignment samples. To address this, we propose splitting evaluation datasets into In-Pretraining (IP) and Out-of-Pretraining (OOP) subsets. This provides a clearer picture of where DG methods succeed and where they fail. We hope that our proposed DomainBed-IP/OOP splits will guide the development of future methods that are better equipped to handle low-alignment data while maintaining performance on high-alignment samples.

## 6 CONCLUSION

We systematically explore how Domain Generalization (DG) methods rely on pre-trained feature alignment from models like CLIP. We hypothesize that the alignment between image and text embeddings during pre-training strongly predicts DG performance. Our experiments confirm this, showing that methods perform well on high-alignment samples (DomainBed-IP) but struggle on low-alignment data (DomainBed-OOP). Notably, state-of-the-art methods like CLIPood perform near oracle-level on aligned data but see significant drops on misaligned samples. This suggests current DG methods rely on pre-trained features and fail to learn new, generalizable features from source domains. Moving forward, two paths emerge: developing DG methods that better learn generalizable features, or focusing on improving pre-trained backbones. While foundation models will continue to advance, there will always be specialized distributions where they fail. Therefore we take the stance that improved DG finetuning remains an important avenure of research. We hope these findings inspire further research into improving generalization on low-alignment data, pushing DG beyond reliance on pre-trained alignment.

## 7 ACKNOWLEDGMENTS

DISTRIBUTION STATEMENT A. Approved for public release. Distribution is unlimited. This material is based upon work supported by the Under Secretary of Defense for Research and Engineering under Air Force Contract No. FA8702-15-D-0001. Any opinions, findings, conclusions or recommendations expressed in this material are those of the author(s) and do not necessarily reflect the views of the Under Secretary of Defense for Research and Engineering.

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

## A APPENDIX

### A.1 TRAINING AND EVALUATION DETAILS

We use a slightly modified MIRO (Cha et al., 2022) codebase for training and evaluation. We use **leave-one-out** evaluation, where a model is trained on all domains except the evaluation domain. We emphasize that we use DomainBed-IP and DomainBed-OOP as evaluation data only, models are trained on full datasets.

For training, we use an OpenCLIP-ViT-B/16 (Ilharco et al., 2021) trained on LAION-400M (Schuhmann et al., 2021). We use default hyper-parameters as defined by (Cha et al., 2022). This includes a learning rate of 5e-5, weight decay of 0.0, a batch size of 32 per-domain, an Adam Optimizer, and no dropout for all methods.

For evaluation, unlike DomainBed, we consider the entire test domain instead of an 80% random split. Following standard practice, we first compute accuracy for each domain, then average those accuracies to get dataset level statistics, and finally compute overall averages averaging across datasets.

For benchmarked methods, we also use hyper-parameters found to be best in respective papers. For **SWAD**, we use an optimum patience parameter value of 3, overfit patience parameter value of 6, and tolerance ratio of 6. For **MIRO**, we use use regularizer loss weight of 1.0. For **CORAL**, we use a CORAL regularizer weight of 1.0, following (Cha et al., 2021). For **LP-FT**, we train the linear probe for 600 steps before unlocking the full backbone. For **Model Parameter Averaging**, we burn in the training for 600 steps before averaging iterates. For **VL2V-SD** and **CLIPood**, we directly use the author's implementation and hyper-parameters, except initializing with OpenCLIP (Ilharco et al., 2021).

### A.2 TRAINING COMPUTE

Each run uses an A6000 48GB GPU, trained for up to 12 hours per domain-dataset combination.

## B ADDITIONAL RESULTS

### B.1 ALIGNMENTSCORE VS ACCURACY

In Figure 11, we plot all benchmarked methods from the main paper, with x-axis corresponding to AlignmentScore, and the y-axis corresponding to the Top-1 Accuracy. We normalize for dataset size, so that no dataset dominates the count. In Figures 12 through 16, we plot these statistics independently per dataset, and find the trends consistent across datasets.

### B.2 PER-DATASET BENCHMARKING RESULTS

We expand Table 1 in the main paper into per-dataset results in Table 10 through 24.

### B.3 SIMILARITY OF TARGET TO PRE-TRAINING

To evaluate the Image Similarity Hypothesis, we retrieve the nearest neighbors from the Laion-400M dataset (Schuhmann et al., 2021). This raises the question of how similar the target domains are to

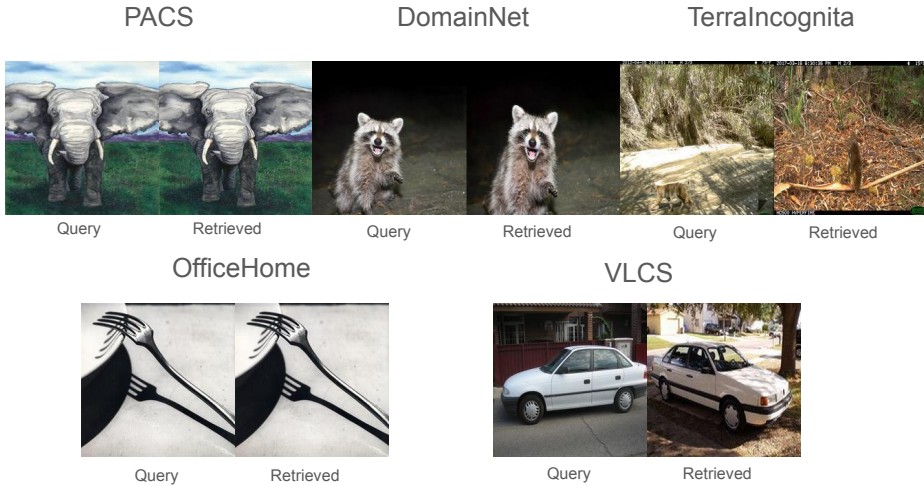

Figure 4: Nearest neighbors of target images in pre-training LAION data.

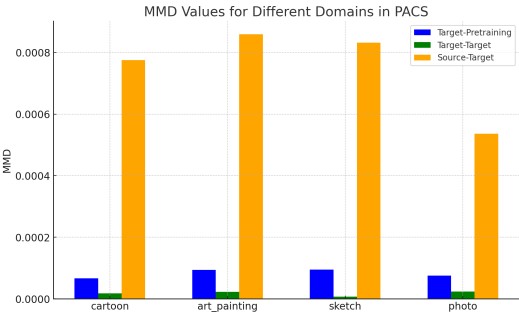

Figure 5: MMD between pre-training and target, source and target, and target and target for PACS. Target is more similar to pre-training than source. Despite this, Alignment is a better predictor of DG performance than perceptual similarity.

the pre-training data and whether the source domains might be even more similar. To investigate this, we compute Maximum Mean Discrepancy (MMD) distances between PACS domains and their nearest neighbors from Laion-400M, as shown in Figure 5. Our results indicate that target domains are, in fact, more similar to the pre-training data than source domains. We inspect nearest neighbors manually, and find even exact duplicates (Figure 4). Interestingly, while we found not only domain-level duplicates but also exact matches in the pre-training data, the Image Similarity Hypothesis is ultimately less predictive than the Alignment Hypothesis.

## B.4 OTHER BACKBONES

We benchmark 2 additional backbones(DINOv2 (Oquab et al., 2023) and OpenAI CLIP) using the MIRO + MPA Domain Generalization method, which we found to be the strongest in our paper, on two datasets (OfficeHome and PACS). This consistency is likely due to the similar nature of the pre-training datasets, both sourced from web scraping and of comparable scale. These findings reinforce that the usefulness of the DomainBed-IP/OOP split is not confined to a specific backbone.

Table 2: Benchmark results of different backbones on OfficeHome and PACS datasets.

| Backbone | OfficeHome - IP | PACS - IP | OfficeHome - OOP | PACS - OOP |
|---|---|---|---|---|
| OpenCLIP-ViT/B 16 | 90.7 | 98.1 | 60.0 | 87.8 |
| CLIP-ViT/B 16 | 88.1 | 97.6 | 57.0 | 86.9 |
| DinoV2 | 87.4 | 97.1 | 58.9 | 85.0 |

Table 3: Splitting DomainBed by PerceptualSimilarityScore. The differences between IP and OOP with this split are much lower than with Alignment Score.

| Dataset | PACS | VLCS | TerraIncognita | OfficeHome | DomainNet |
|---|---|---|---|---|---|
| Perceptual IP | 97.2 | 74.8 | 60.6 | 86.8 | 61.4 |
| Perceptual OOP | 95.6 | 77.4 | 42.9 | 78.3 | 55.3 |

## B.5 Splitting DomainBed using PerceptualSimilarityScore

In Figure 2 a.), we show that the slope of the relationship of Top-1 Accuracy vs Perceptual Similarity Score is positive but shallow. This suggests that using PerceptualSimilarityScore as an alternative to AlignmentScore for splitting DomainBed would not be very effective. To further prove this point, we split at a PerceptualSimilarityScore value of 0.86 in Table 3. We can see the differences are not very large between OOP and IP, indicating that AlignmentScore is a better thresholding metric.

## B.6 Comparing AlignmentScore with zero-shot classification confidence score

We also consider an alignment score which takes into account uncertainties, and compute a score using the confidence of the zero-shot classifier formed by the pre-trained CLIP model for each sample. Specifically, for a sample with ground truth class c, we calculate the softmax over the logits output by the zero-shot classifier, and use the resulting probability p(c) as the score. We refer to this as the Calibrated AlignmentScore and show the results in Figure 6. Although the score predicts generalization for both OfficeHome and DomainNet, the scores have different scales for different datasets. In contrast out AlignmentScore does align across datasets to a greater degree (Figure 7)

We explore the effect of averaging PerceptualSimilarityScore and AlignmentScore in Figure 8. We can see that there is not much of a compositional effect, so therefore we stick with AlignmentScore as our generalization predictor.

## B.7 Image Similarity Hypothesis for source data

The main drawback of the Image Similarity hypothesis is that it does not consider how well the nearest perceptual neighbor is learned during pre-training. One reason for a sample being poorly learned during pre-training is that the pre-training caption is not very relevant to the DG task. Source data is unlikely to have this issue, since source and target domains share labels. Therefore it is interesting to ask how strongly correlated the PerceptualSimilarityScore is with DG accuracy when measured between source and target. Indeed , as seen in Figure 9, there is a strong correlation. However, simply using source-data to compute the PerceptualSimilarity results in an incomplete understanding of the relationship between target data and the training procedure, due to the lack of consideration of the pre-training. In fact, zero-shot models with NO learning from source are very performant (Table 1)

## B.8 Training from scratch

We focus on analyzing the generalization capabilities of CLIP-based DG methods, as these have been demonstrated to be the strongest in recent work. However,from-scratch experiments are valuable to measure how much finetuning on source data can learn, and so we also trained ResNet-50 models from scratch, using the MPA (Model Parameter Averaging) method and present the results in Table 4. We can see that performance is very low even on ALL DomainBed data (unsplit), further

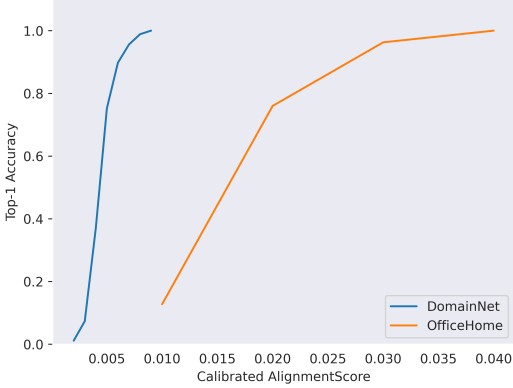

Figure 6: Top-1 DG Accuracy vs calibrated alignment: We use the confidence of the zero-shot classifier formed by the pre-trained CLIP models an alignment measure. Although the score predicts generalization for both OfficeHome and DomainNet, the scores have different scales for different datasets.

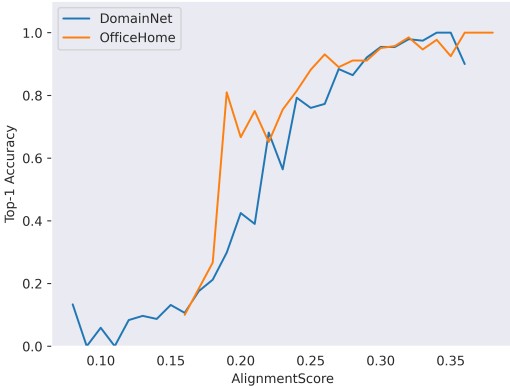

Figure 7: Top-1 DG Accuracy vs AlignmentScore: The AlignmentScore introduced in our work have scales which are comparable across different datasets.

confirming that existing DG algorithms are poor at learning new knowledge from source data alone, and they rely on strong pre-training for good performance.

### B.9 ANALYSIS OF ZERO-SHOT BEHAVIOR

To better understand the behavior of Domain Generalization (DG) methods, we compute confusion matrices for zero-shot (ZS) and In-Pretraining (IP) and Out-of-Pretraining (OOP) subsets for the PACS dataset using CLIPood. The results are shown in Tables 5 and 6.

The confusion matrices reveal key insights about the behavior of DG methods:

- For the **IP subset**, DG methods catastrophically forget a significant number of samples, flipping correct predictions made by zero-shot models to incorrect ones. This suggests that DG methods can even have a *negative value* for IP data. Nevertheless, performance remains strong.

- For the **OOP subset**, DG methods flip very few correct samples to incorrect ones. However, the state-of-the-art (SOTA) method, CLIPood, is still unable to correct the majority of incorrectly classified samples in this subset.

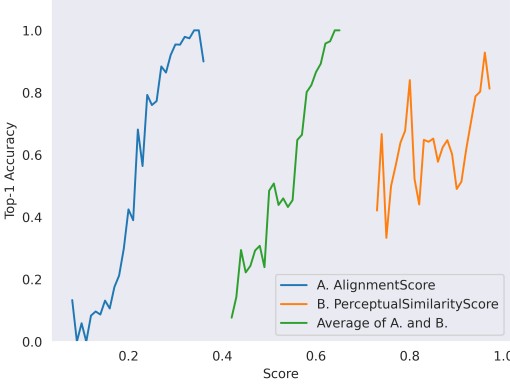

Figure 8: Combining PerceptualSimilarity Score and AlignmentScore: We explore the effect of averaging PerceptualSimilarityScore and AlignmentScore. There is no visible additional signal from averaging. We therefore stick with AlignmentScore as our generalization predictor.

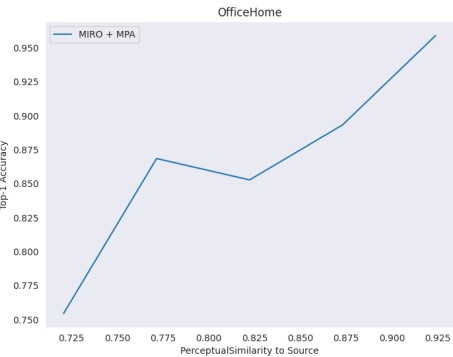

Figure 9: PerceptualSimilarity to Source for OfficeHome. DG performance is correlated with similarity to source images.

Overall, in both IP and OOP cases, the results indicate that CLIPood and other DG methods rely heavily on pre-training. These findings underscore the limitations of current DG methods and highlight areas for improvement.

## C ADDITIONAL DOMAINBED-(IP/OOP) STATISTICS AND ANALYSIS

### C.1 CONSTITUENT DATASETS

We split 5 datasets in DomainBed-(IP/OOP),( VLCS (Fang et al., 2013), DomainNet (Peng et al., 2019), OfficeHome (Ganin et al., 2016), PACS (Li et al., 2017), and TerraIncognita (Beery et al., 2018)). Here we provide basic statistics of each.

**VLCS** has 5 classes and 4 domains: Caltech101, LabelME, SUN09, and VOC2007, with 10729 samples. The domain shift is dataset source.

**DomainNet** contrains 345 classes and 6 domains: clipart, infograph, quickdraw, real, and sketch. It has a total of 586,575 samples. The dataset shift is style.

**OfficeHome** has 65 classes and 4 domains: art, clipart, product, and real. The dataset shift is style.

**TerraIncognita** has 10 classes of wildlife cameras. There are 4 domains of different cameras and 24788 samples. The dataset shift is camera location.

Table 4: Training ResNet-50 from scratch. DG performance is very poor, showing DG methods rely on strong pre-training

|  | DomainNet | OfficeHome | PACS | TI | VLCS | Avg |
|---|---|---|---|---|---|---|
| ResNet-50 from Scratch | 7.6 | 17.2 | 36.3 | 23.8 | 55.2 | 28.0 |
| Clip ViT-B/16 | 58.9 | 82.0 | 94.3 | 50.7 | 82.3 | 73.6 |

Table 5: Confusion matrix for the IP subset of the PACS dataset.

|  | **DG: Incorrect** | **DG: Correct** |
|---|---|---|
| **ZS: Incorrect** | 141 | 70 |
| **ZS: Correct** | 179 | 8231 |

**PACS** has 9991 sampes and 4 domains: arts, cartoon, photo, and sketch. There are 7 classes. The dataset shift is style.

## C.2    MISLABELING AND OCR RATES

In Table 7 we present the mislabelling rates at various AlignmentScore values. In Table 8, we present how much data is removed due to mislabelling or OCR filtering.

## C.3    CLASS DISTRIBUTION OF DOMAINBED-(IP/OOP):

In Figures 19 through 33, we provide class distribution statistics of different datasets before splitting and in our IP and OOP splits. We find some interesting patterns. For example, in Office-Home, the OOP class is dominated by **marker** and **toys**, while the IP split has a much more uniform distribution. Similarly, both PACS (DomainBed-OOP) and VLCS (DomainBed-OOP) are dominated by **person**.

## C.4    VISDIFF DIFFERENCES BETWEEN DB-IP AND DB-OOP

We further investigate the distinctions between the IP and OOP subsets using VisDiff (Dunlap et al., 2024), an LLM-based system that identifies differences between sets of images. For each combination of dataset, domain, and class—except for DomainNet, where we subsample roughly 15% of the combinations due to computational constraints—we independently sample up to 30 images from both the IP and OOP subsets. VisDiff employs CLIP to compute an AUC-ROC score for the natural language differences proposed by the LLM, and we retain only those differences with an AUC score of 0.7 or higher. The complete results are presented in Table 9. We find that contextual or environmental elements often overshadow the primary object indicated by the text label. For example, in the VLCS dataset's SUN09 domain, the OOP subset for the car class frequently features images described as "historical architecture," while in the Office Home dataset, real-domain bed images are deemed OOP if they include "child-themed decor."

## C.5    RELATIONSHIP BETWEEN OBJECT SIZE AND ALIGNMENTSCORE

To test whether object size is correlated with AlignmentScore, we computed the object size of the ground truth labeled objects using bounding boxes generated by the open-world object detector OWLv2 (Minderer et al., 2023). Overall, object size is correlated with the AlignmentScore (see Figure 10), with the sole exception being the TerraIncognita dataset. This indicates that even the presence of small wildlife can substantially increase the AlignmentScore, and that the background vegetation does not present a significantly conflicting signal that is sometimes present in other datasets. Overall, this suggests that pre-training which better represents scenes with multiple objects or concepts may improve benchmark performance.

Table 6: Confusion matrix for the OOP subset of the PACS dataset.

|                | DG: Incorrect | DG: Correct |
|----------------|---------------|-------------|
| **ZS: Incorrect** | 98         | 62          |
| **ZS: Correct**   | 15         | 890         |

Table 7: Mislabeling rates across different AlignmentScore ranges.

| Dataset        | 0.0-0.05 | 0.05-0.10 | 0.10-0.15 | 0.15-0.20 |
|----------------|----------|-----------|-----------|-----------|
| OfficeHome     | 100%     | 45%       | 28%       | 12%       |
| PACS           | -        | 46%       | 5%        | 0%        |
| TerraIncognita | -        | 55%       | 34%       | 23%       |
| DomainNet      | 65%      | 35%       | 33%       | 11%       |
| VLCS           | 33%      | 14%       | 9%        | 3%        |

Table 8: Percentage of discarded samples due to mislabeling or label text in the image (OCR).

| Dataset        | Dropped - Mislabelled | Dropped - OCR |
|----------------|-----------------------|---------------|
| OfficeHome     | 0.92%                 | 0.15%         |
| PACS           | 3.05%                 | 0.00%         |
| TerraIncognita | 0.22%                 | 0.00%         |
| DomainNet      | 7.64%                 | 0.03%         |
| VLCS           | 12.41%                | 0.00%         |

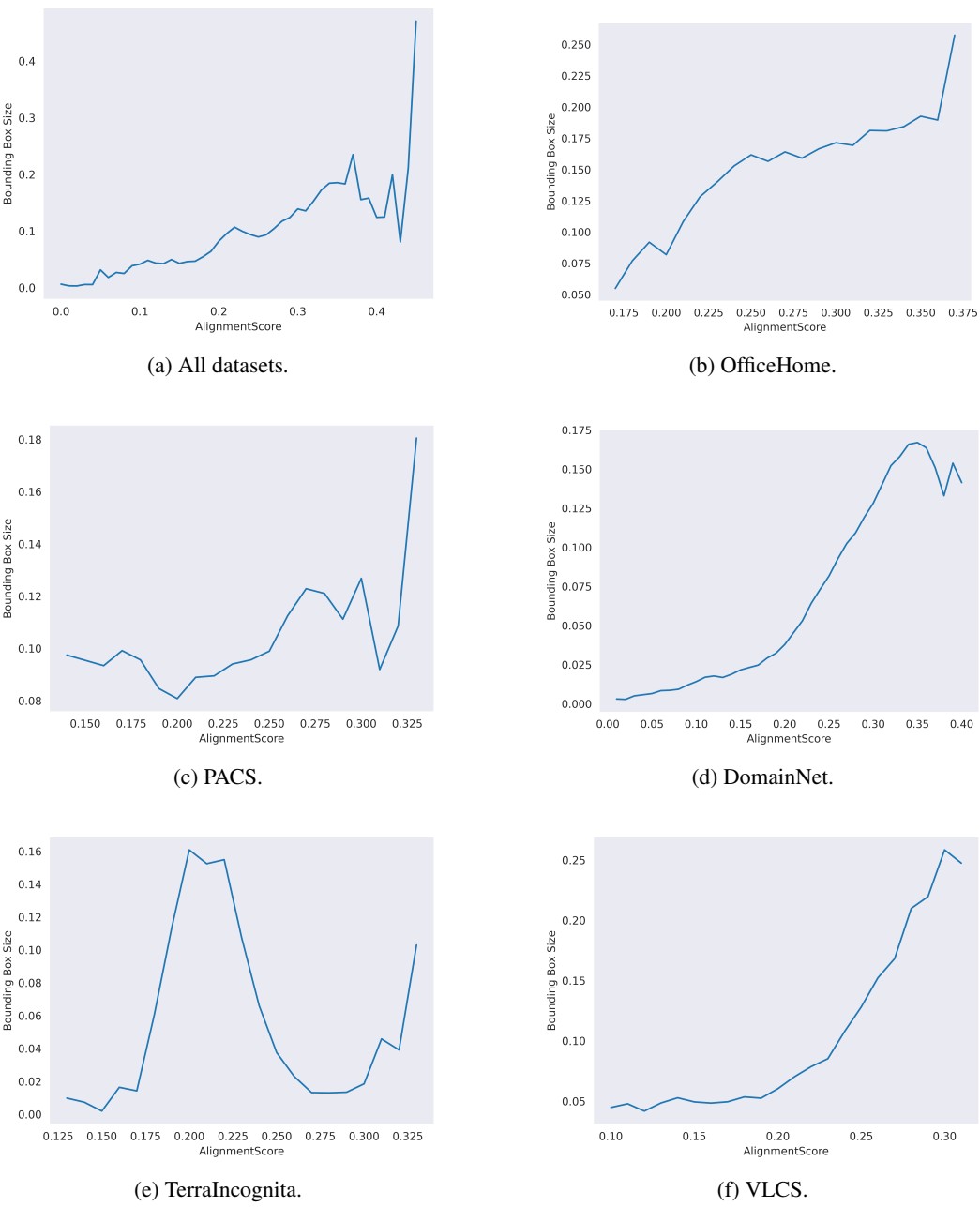

Figure 10: Overall relationship between ground truth object bounding box size and AlignmentScore across different DomainBed datasets. Overall the relationship is positive, indicating that object size plays a role in whether a data point is IP or OOP. The exception is TerraIncognita, indicating that camera trap backgrounds do not strongly conflict with the presence of the wildlife to be classified. See Section C.5.

Table 9: VisDiff: Differences Identified for Image Sets. We use VisDiff (Dunlap et al., 2024), an LLM-based system for describing differences between images sets, to identify differences in between IP and OOP images. We do so independently for each dataset, domain, and class grouping. Distractors unrelated to the class seem to be an issue. For more details see Section C.4.

| Dataset_Domain_Class | Description of OOP Subset |
|---|---|
| VLCS_SUN09_car | Historical architecture |
| VLCS_SUN09_bird | variety of landscapes |
| VLCS_SUN09_person | indoor water parks |
| VLCS_Caltech101_dog | text about adoption and rescue services |
| VLCS_LabelMe_chair | people walking on city streets |
| VLCS_LabelMe_dog | gatherings of people |
| VLCS_Caltech101_bird | paintings |
| VLCS_Caltech101_person | murals on buildings |
| VLCS_VOC2007_dog | inside homes with multiple people |
| VLCS_Caltech101_car | presence of branded vehicles |
| VLCS_SUN09_dog | cooking areas |
| VLCS_VOC2007_car | double decker buses |
| VLCS_LabelMe_bird | people near bodies of water |
| VLCS_VOC2007_person | equestrian activities |
| VLCS_VOC2007_bird | people engaging in activities |
| terra_incognita_location_38_rabbit | sunny daytime scenes |
| terra_incognita_location_38_squirrel | streams in wooded areas |
| terra_incognita_location_46_opossum | animal walking |
| terra_incognita_location_100_dog | muddy areas |
| terra_incognita_location_46_raccoon | cloud formations |
| terra_incognita_location_43_dog | streams in the forest |
| terra_incognita_location_100_bird | a dirt road in a garden |
| terra_incognita_location_46_squirrel | camera trap captures |
| terra_incognita_location_38_bird | night vision |
| terra_incognita_location_46_empty | middle of the night |
| terra_incognita_location_100_rabbit | absence of man-made objects |
| terra_incognita_location_38_opossum | screens with green backgrounds |
| terra_incognita_location_46_dog | crocodiles |
| terra_incognita_location_46_rabbit | shadows in nature |
| terra_incognita_location_100_squirrel | dirt roads |
| terra_incognita_location_38_cat | a black bear resting on a log |
| terra_incognita_location_43_squirrel | muddy areas |
| terra_incognita_location_43_bobcat | an image of a crocodile |
| terra_incognita_location_43_bird | deserts |
| terra_incognita_location_38_coyote | black bears |
| terra_incognita_location_46_bobcat | streams in a wooded area |
| terra_incognita_location_46_coyote | waterfalls |
| terra_incognita_location_46_cat | Wooded areas |
| terra_incognita_location_43_coyote | crocodiles |
| terra_incognita_location_43_opossum | boats in water |
| terra_incognita_location_100_cat | images of tapirs at night |
| terra_incognita_location_100_raccoon | natural landscape |
| terra_incognita_location_43_cat | animals resting |
| terra_incognita_location_43_raccoon | wildlife in water |
| office_home_Art_Paper_Clip | animals hanging |
| office_home_Clipart_Webcam | colorful backgrounds |
| office_home_Product_Soda | rows of multiple beverage containers |
| office_home_Real_Speaker | jbl portable bluetooth speakers |
| office_home_Product_Drill | yellow and black color scheme |
| office_home_Real_Clipboards | advertising signage |

Table 9

| Dataset_Domain_Class | Description of OOP Subset |
|---|---|
| office_home_Clipart_Fan | close-up views |
| office_home_Real_Bike | road cycling |
| office_home_Product_Lamp_Shade | circular objects |
| office_home_Clipart_Notebook | textbooks with colorful covers |
| office_home_Real_Sink | brushed nickel finish |
| office_home_Clipart_Keyboard | singular computer components |
| office_home_Art_Speaker | toy-like objects |
| office_home_Product_Radio | bluetooth devices |
| office_home_Real_Glasses | space-related elements |
| office_home_Clipart_Desk_Lamp | asian clothing styles |
| office_home_Art_Bottle | earrings with a bottle and cork |
| office_home_Clipart_Clipboards | a promotional offer |
| office_home_Real_Flowers | butterflies |
| office_home_Product_Fan | air purifiers |
| office_home_Clipart_Alarm_Clock | cartoon characters |
| office_home_Art_Backpack | toys with wings |
| office_home_Art_Computer | illustrations |
| office_home_Art_Shelf | flowers in pots |
| office_home_Product_Pen | focus on packaging |
| office_home_Product_Mouse | colorful electronics on display |
| office_home_Clipart_Postit_Notes | notebooks on a table |
| office_home_Product_Clipboards | natural materials without additional items |
| office_home_Product_Batteries | set of six objects |
| office_home_Clipart_Curtains | feminine imagery |
| office_home_Art_Pan | eggs with faces drawn on them |
| office_home_Clipart_Telephone | 3D rendering |
| office_home_Clipart_Monitor | technology themes |
| office_home_Product_Screwdriver | close-up of a person engaged in an activity |
| office_home_Clipart_Paper_Clip | Red color prominence |
| office_home_Product_Calendar | octagon shapes |
| office_home_Product_Couch | close-up furniture shots |
| office_home_Art_Helmet | dragon's head |
| office_home_Clipart_File_Cabinet | 3D rendering |
| office_home_Clipart_Shelf | commercial products |
| office_home_Clipart_TV | realistic images |
| office_home_Real_Curtains | a bathroom setting |
| office_home_Art_Toys | a small white dog |
| office_home_Art_Glasses | individual with headgear |
| office_home_Product_Sneakers | gel cushioning |
| office_home_Real_Drill | blue colored objects |
| office_home_Clipart_Glasses | audio equipment |
| office_home_Clipart_Lamp_Shade | black and white drawing |
| office_home_Real_Bed | child-themed decor |
| office_home_Art_File_Cabinet | people in a professional setting |
| office_home_Product_Backpack | focus on material texture |
| office_home_Art_Printer | office settings with unusual elements |
| office_home_Real_Batteries | emphasis on a specific brand |
| office_home_Art_Soda | cans of angry birds tropic soda |
| office_home_Real_Trash_Can | objects on wheels |
| office_home_Product_Eraser | packages explicitly labeled 'pentel' |
| office_home_Real_Shelf | laundry room |
| office_home_Clipart_Eraser | book on a black background |
| office_home_Clipart_ToothBrush | distinctive backgrounds |
| office_home_Product_Glasses | sunglasses with red lenses |
| office_home_Clipart_Calendar | sanitation theme |

Table 9

| Dataset_Domain_Class | Description of OOP Subset |
|---|---|
| office_home_Clipart_Mop | detailed human figures |
| office_home_Art_Oven | combination of domestic and clinical spaces |
| office_home_Real_Lamp_Shade | minimalist style |
| office_home_Art_Fan | digital artwork |
| office_home_Art_Webcam | futuristic design |
| office_home_Art_Laptop | fantasy or fictional themes |
| office_home_Art_Bed | enchantment |
| office_home_Art_Curtains | dark hair |
| office_home_Real_Fan | multiple ceiling fans |
| office_home_Clipart_Pen | cocktail shakers |
| office_home_Art_Screwdriver | blue and silver color palette |
| office_home_Product_Keyboard | gaming mouse included |
| office_home_Real_Notebook | colorful stationery |
| office_home_Real_Couch | contemporary living spaces |
| office_home_Art_Table | standing posture |
| office_home_Art_Pen | creative process |
| office_home_Real_Mop | clothes hanging |
| office_home_Art_Scissors | video game characters |
| office_home_Clipart_Scissors | abstract shapes |
| office_home_Art_Keyboard | concrete structures |
| office_home_Real_Calendar | printable charts with decorative backgrounds |
| office_home_Art_Calculator | decorative elements |
| office_home_Art_Bike | animal on a vehicle |
| office_home_Product_Webcam | a single computer monitor |
| office_home_Clipart_Bottle | household container |
| office_home_Clipart_Computer | a printer |
| office_home_Real_Sneakers | Black high top sneakers |
| office_home_Clipart_Mouse | monochromatic themes |
| office_home_Art_Notebook | objects with a vintage look |
| office_home_Clipart_Bucket | objects associated with liquids |
| office_home_Product_Laptop | abstract simplicity |
| office_home_Art_Push_Pin | digital art |
| office_home_Real_Push_Pin | objects placed against natural backgrounds |
| office_home_Art_Ruler | animal skulls |
| office_home_Art_Couch | cats |
| office_home_Art_Desk_Lamp | intricate costumes |
| office_home_Art_Mouse | unique artistic interpretations |
| office_home_Art_Refrigerator | a toy girl in an unusual setting |
| office_home_Product_Knives | survival or tactical items |
| office_home_Product_Helmet | an image of a protective gear |
| office_home_Clipart_Helmet | abstract concepts |
| office_home_Product_Notebook | calendar |
| office_home_Product_Paper_Clip | a caliper and a paper clip |
| office_home_Product_Hammer | hammer head detail |
| office_home_Clipart_Bed | detailed human emotions |
| office_home_Clipart_Chair | a red background |
| office_home_Art_Flowers | cakes with unique decorations |
| office_home_Product_Toys | White and brown plush toys |
| office_home_Art_Fork | red dress |
| office_home_Real_Pan | cooked fish |
| office_home_Real_Bucket | tabletop settings |
| office_home_Art_Trash_Can | drinks on a colorful background |
| office_home_Real_Knives | focus on object |
| office_home_Real_Pencil | pink color theme |
| office_home_Product_File_Cabinet | green objects on a white background |

Table 9

| Dataset_Domain_Class | Description of OOP Subset |
|---|---|
| office_home_Real_Hammer | German craftsmanship |
| office_home_Art_Sink | illustrations |
| office_home_Clipart_Screwdriver | minimalistic design |
| office_home_Art_Marker | illustrations of animals |
| office_home_Art_Telephone | sport equipment |
| office_home_Real_Folder | blue wall |
| office_home_Product_Folder | focus on texture |
| office_home_Clipart_Batteries | objects on a plain, uncluttered background |
| office_home_Product_Mug | black handle |
| office_home_Product_Flowers | a single red rose |
| office_home_Real_Soda | mango juice with a straw |
| office_home_Art_Lamp_Shade | people with hats |
| office_home_Clipart_Oven | focus on a single object |
| office_home_Art_Spoon | fantasy elements |
| office_home_Clipart_Table | cartoons |
| office_home_Clipart_Hammer | food items |
| office_home_Art_Radio | yellow boomboxes |
| office_home_Clipart_Sink | human activity |
| office_home_Clipart_Drill | digital art style |
| office_home_Art_Eraser | illustrations or drawings |
| office_home_Clipart_Exit_Sign | clear and singular message |
| office_home_Clipart_Pencil | a spool of ribbon |
| office_home_Art_Batteries | graphic t-shirts |
| office_home_Art_Mug | coffee with food items inside |
| office_home_Clipart_Bike | signs with text |
| office_home_Art_TV | multiple tv screens |
| office_home_Product_Chair | inflatable furniture |
| office_home_Art_Drill | conceptual illustrations |
| office_home_Art_Postit_Notes | cookies |
| office_home_Art_Chair | fire or flames |
| office_home_Art_Hammer | women in fantasy attire |
| office_home_Art_Folder | men in suits |
| office_home_Product_Computer | MSI branded devices |
| office_home_Product_Push_Pin | geographic representations |
| office_home_Clipart_Laptop | a pair of devices |
| office_home_Art_Sneakers | octopuses on shoes |
| office_home_Art_Candles | whimsical objects |
| office_home_Real_Chair | ottomans |
| office_home_Clipart_Pan | stacked objects |
| office_home_Art_Bucket | man playing music |
| PACS_cartoon_person | images featuring ice cream |
| PACS_art_painting_dog | hunting scenes |
| PACS_art_painting_giraffe | deer in the artwork |
| PACS_art_painting_guitar | artwork displaying paint splatters |
| PACS_art_painting_person | artistic depictions featuring abstract or imaginative elements |
| PACS_cartoon_horse | giraffes |
| PACS_cartoon_guitar | hand-drawn illustrations |
| PACS_photo_dog | dogs in vehicles |
| PACS_photo_horse | non-living objects |
| PACS_art_painting_horse | bulls |
| domain_net_clipart_shorts | characters with speech bubbles |
| domain_net_infograph_bottlecap | technical illustrations |
| domain_net_infograph_kangaroo | different types of fish in infographics |

Table 9

| Dataset_Domain_Class | Description of OOP Subset |
|---|---|
| `domain_net_clipart_bicycle` | scenes involving flight |
| `domain_net_sketch_sock` | monkeys as a subject |
| `domain_net_painting_stethoscope` | text with humor or messages |
| `domain_net_infograph_aircraft_carrier` | an organized visual representation of navy ships |
| `domain_net_clipart_The_Eiffel_Tower` | black background |
| `domain_net_sketch_submarine` | educational elements like drawing guides |
| `domain_net_clipart_parrot` | blue birds |
| `domain_net_painting_string_bean` | paintings on a striped background |
| `domain_net_real_house` | large resorts |
| `domain_net_real_stop_sign` | images near a river |
| `domain_net_clipart_mouse` | cute small animals |
| `domain_net_real_hedgehog` | objects in a grassy area |
| `domain_net_real_cat` | presence of dogs |
| `domain_net_painting_bucket` | watercolor paintings |
| `domain_net_sketch_motorbike` | steampunk design |
| `domain_net_sketch_wine_glass` | cocktails |
| `domain_net_painting_strawberry` | lace tablecloths |
| `domain_net_sketch_bus` | text elements in illustrations |
| `domain_net_infograph_flower` | floral arrangements |
| `domain_net_clipart_tiger` | Airplanes flying in the sky |
| `domain_net_clipart_lobster` | repeated phrases or words |
| `domain_net_sketch_baseball` | a basketball on a wooden floor |
| `domain_net_infograph_steak` | comparison charts |
| `domain_net_sketch_pillow` | teddy bear motifs |
| `domain_net_clipart_fish` | black and white illustrations |
| `domain_net_infograph_hockey_puck` | infographics focusing on player statistics |
| `domain_net_painting_bandage` | children in art |
| `domain_net_painting_harp` | circular tables |
| `domain_net_sketch_ant` | clouds |
| `domain_net_sketch_calculator` | squares and shapes |
| `domain_net_sketch_church` | specific locations |
| `domain_net_infograph_paintbrush` | informative visual aids |
| `domain_net_painting_sea_turtle` | stained glass |
| `domain_net_clipart_candle` | birthday-themed images |
| `domain_net_clipart_baseball_bat` | characters with beverages |
| `domain_net_painting_squirrel` | a combination of different artistic styles |
| `domain_net_infograph_trumpet` | mushrooms |
| `domain_net_real_cloud` | illustrations of devices |
| `domain_net_infograph_string_bean` | dairy farm flyer |
| `domain_net_clipart_scorpion` | designs for personal adornment |
| `domain_net_painting_mouth` | art creation process |
| `domain_net_painting_pickup_truck` | people creating art |
| `domain_net_painting_scorpion` | paintings with multiple subjects |
| `domain_net_clipart_asparagus` | chef's hat |
| `domain_net_real_elephant` | frogs |
| `domain_net_painting_suitcase` | scenes involving rivers or water |
| `domain_net_real_tornado` | a video game setting |
| `domain_net_painting_pond` | a painting of people |
| `domain_net_painting_oven` | indoor car scenes |
| `domain_net_infograph_flip_flops` | modern info presentation template |
| `domain_net_clipart_vase` | casual outdoors |
| `domain_net_clipart_dolphin` | multiple animal species |
| `domain_net_real_lightning` | people indoors |
| `domain_net_infograph_banana` | recipes |

Table 9

| Dataset_Domain_Class | Description of OOP Subset |
|---|---|
| domain_net_painting_cooler | incomplete words on objects |
| domain_net_sketch_tractor | depiction of people |
| domain_net_clipart_power_outlet | colorful graphic design |
| domain_net_infograph_bandage | infographics |
| domain_net_real_string_bean | recipes with bacon |
| domain_net_infograph_sheep | leather industry |
| domain_net_real_carrot | hummus |
| domain_net_painting_lighthouse | murals |
| domain_net_clipart_skyscraper | flyer designs |
| domain_net_sketch_blueberry | a set of hand drawn fruit |
| domain_net_sketch_snorkel | cartoon animals |
| domain_net_infograph_vase | flyers with promotional content |
| domain_net_real_moon | mythological or fantasy elements |
| domain_net_sketch_hamburger | fast food with ice cream |
| domain_net_infograph_barn | organic farming elements |
| domain_net_clipart_rainbow | umbrellas |
| domain_net_clipart_dog | images featuring human characters |
| domain_net_infograph_elbow | educational posters |
| domain_net_clipart_face | vector illustrations |
| domain_net_sketch_hourglass | vector illustrations |
| domain_net_sketch_laptop | people sitting at desks |
| domain_net_real_fire_hydrant | airplane in the sky |
| domain_net_real_mouse | a cable connection |
| domain_net_real_hourglass | jewelry |
| domain_net_painting_whale | paintings in a gallery setting |
| domain_net_painting_envelope | mixed media paintings |
| domain_net_sketch_flip_flops | illustrations of hats and sunglasses |
| domain_net_infograph_hedgehog | infographics with statistical data |
| domain_net_painting_sun | depictions of solitude |
| domain_net_painting_knee | military uniforms |
| domain_net_clipart_butterfly | birds and flowers |
| domain_net_sketch_goatee | illustrations of chefs |
| domain_net_real_airplane | dynamic outdoor sports |
| domain_net_sketch_banana | hand drawn illustrations |
| domain_net_infograph_lollipop | sweet-themed infographics |
| domain_net_real_owl | colorful designs |
| domain_net_real_banana | plate presentations |
| domain_net_painting_tornado | people in colorful dresses |
| domain_net_real_lobster | multiple dishes presented together |
| domain_net_sketch_frog | people drawing with different tools |
| domain_net_real_mountain | extreme sports |
| domain_net_real_drums | guitar |
| domain_net_infograph_knee | recovery processes |
| domain_net_real_teapot | coffee mugs |
| domain_net_real_camouflage | images with helicopters |
| domain_net_infograph_parrot | first aid themes |
| domain_net_infograph_bathtub | wheelchair-accessible features |
| domain_net_painting_bracelet | artistic portraits |
| domain_net_painting_fireplace | people sitting together |
| domain_net_sketch_owl | Rough outlines |
| domain_net_sketch_crocodile | lizards and frogs |
| domain_net_clipart_kangaroo | Christmas themes |
| domain_net_clipart_cat | cartoons with both cats and dogs |
| domain_net_real_golf_club | screenshots of video games |
| domain_net_painting_laptop | illustrated people |

Table 9

| Dataset_Domain_Class | Description of OOP Subset |
| --- | --- |
| domain_net_clipart_broom | dog in a witch's hat with a pumpkin |
| domain_net_painting_bulldozer | night scenes indoors |
| domain_net_sketch_sailboat | nautical-themed accessories |
| domain_net_clipart_picture_frame | Colorful paw prints |
| domain_net_clipart_octagon | colorful gumballs |
| domain_net_clipart_ladder | whimsical illustrations |
| domain_net_clipart_scissors | illustrations with human figures |
| domain_net_clipart_tractor | images of airplanes |
| domain_net_real_chandelier | rooms with large beds |
| domain_net_sketch_guitar | Everyday scenes involving musical themes |
| domain_net_clipart_broccoli | promotional style imagery |
| domain_net_clipart_ice_cream | refrigerators |
| domain_net_painting_octagon | birds |
| domain_net_clipart_church | ceremonial events |
| domain_net_infograph_sleeping_bag | child safety |
| domain_net_painting_drill | colorful outdoor scenes |
| domain_net_painting_hot_tub | green character in a bathroom scene |
| domain_net_clipart_rhinoceros | cartoon characters |
| domain_net_sketch_roller_coaster | people on a piece of paper |
| domain_net_painting_bridge | Venice |
| domain_net_clipart_violin | animals playing instruments |
| domain_net_real_mosquito | abstract or surreal elements |
| domain_net_sketch_spider | comic book grading |
| domain_net_infograph_donut | checkered patterns |
| domain_net_painting_stairs | art restoration |
| domain_net_clipart_flamingo | patterns and designs |
| domain_net_real_knee | people in stylish outfits |
| domain_net_infograph_basketball | detailed sports statistics |
| domain_net_clipart_sleeping_bag | colorful designs |
| domain_net_painting_remote_control | watercolor |
| domain_net_painting_calculator | a person multitasking |
| domain_net_real_speedboat | tourist beaches |
| domain_net_clipart_bus | psychedelic imagery |
| domain_net_painting_speedboat | images with a variety of painting mediums |
| domain_net_painting_toilet | decorative art |
| domain_net_painting_cell_phone | colorful designs |
| domain_net_sketch_beard | scenes depicting historical or fictional characters |
| domain_net_clipart_key | a man in a hat |
| domain_net_clipart_spreadsheet | vector illustrations with folders |
| domain_net_infograph_mushroom | identification charts |
| domain_net_clipart_nose | cartoon animal with a flower |
| domain_net_painting_stop_sign | oil paintings |
| domain_net_sketch_baseball_bat | vector illustrations on a black background |
| domain_net_real_elbow | colorful clothing |
| domain_net_infograph_feather | posters for events |
| domain_net_clipart_dragon | an airplane flying |

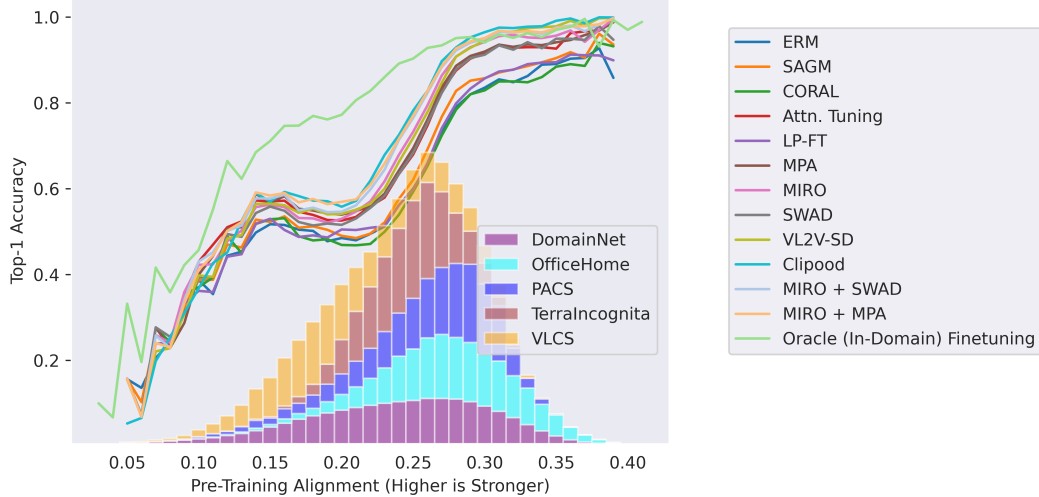

Figure 11: Plotting AlignmentScore vs Top-1 Accuracy for all benchmarked methods, for all datasets together. Although some methods are stronger than others, all follow the same trend of increasing accuracy with increased AlignmentScore.

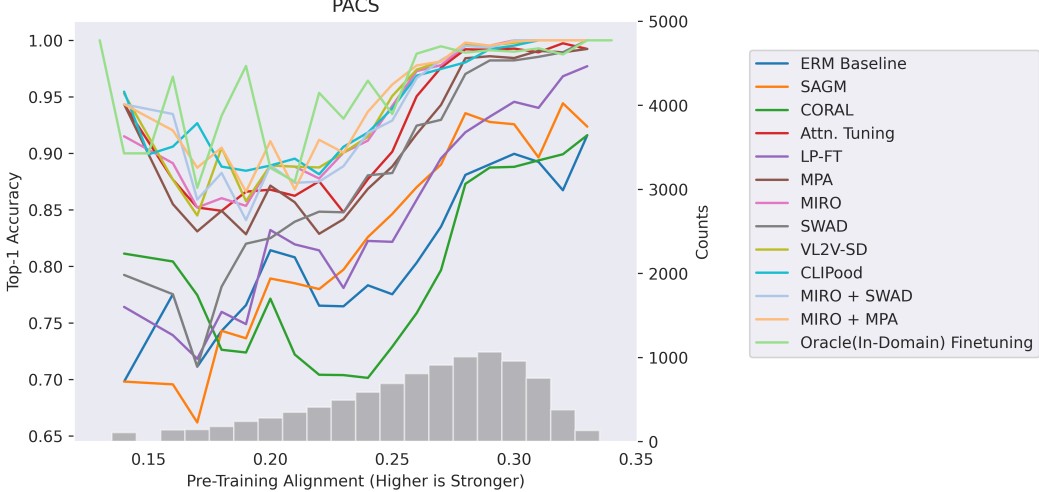

Figure 12: Plotting AlignmentScore vs Top-1 Accuracy on the PACS dataset.

Table 10: Per-domain breakdown for OfficeHome (DomainBed-OOP)

| Method | Art | Clipart | Product | Real | Avg |
|---|---|---|---|---|---|
| OpenClip ZS | 49.4 | 24.6 | 59.0 | 59.3 | 48.1 |
| CORAL | 33.0 | 29.4 | 50.0 | 58.0 | 42.6 |
| SAGM | 36.8 | 30.7 | 51.7 | 58.7 | 44.5 |
| ERM | 36.0 | 28.1 | 51.3 | 56.0 | 42.9 |
| LP-FT | 37.2 | 29.7 | 52.0 | 54.7 | 43.4 |
| SWAD | 43.9 | 37.1 | 56.0 | 62.7 | 49.9 |
| MIRO | 49.4 | 39.6 | 62.7 | 74.7 | 56.6 |
| VL2V-SD | 56.4 | 39.3 | 65.3 | 65.3 | 56.6 |
| Attn. Tune | 46.0 | 36.1 | 59.7 | 64.0 | 51.4 |
| Model Parameter Averaging (MPA) | 46.0 | 36.4 | 55.7 | 66.0 | 51.0 |
| CLIPOOD | 60.3 | 44.4 | 73.3 | 78.0 | 63.9 |
| MIRO + SWAD | 55.2 | 44.7 | 63.3 | 72.7 | 59.0 |
| MIRO + MPA | 53.5 | 44.1 | 65.7 | 76.7 | 60.0 |

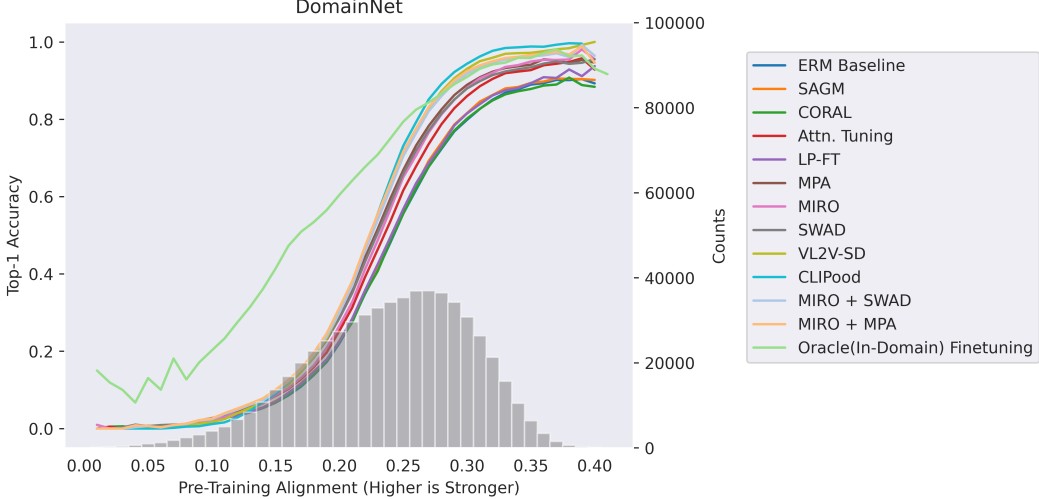

Figure 13: Plotting AlignmentScore vs Top-1 Accuracy on the DomainNet dataset.

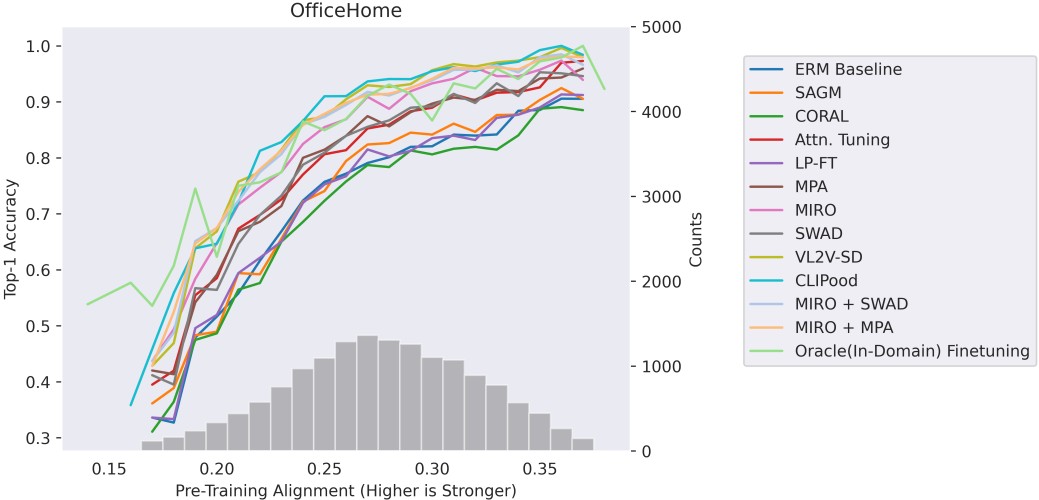

Figure 14: Plotting AlignmentScore vs Top-1 Accuracy on the OfficeHome dataset.

Table 11: Per-domain breakdown for OfficeHome (DomainBed-IP)

| Method | Art | Clipart | Product | Real | Avg |
|---|---|---|---|---|---|
| OpenClip ZS | 88.5 | 78.0 | 95.4 | 93.9 | 88.9 |
| CORAL | 71.6 | 62.9 | 85.8 | 84.0 | 76.1 |
| SAGM | 75.7 | 70.8 | 87.0 | 84.5 | 79.5 |
| ERM* | 74.1 | 67.4 | 86.7 | 84.1 | 78.1 |
| LP-FT | 75.2 | 66.9 | 88.9 | 83.0 | 78.5 |
| SWAD | 81.4 | 77.8 | 91.8 | 88.2 | 84.8 |
| MIRO | 88.3 | 81.1 | 93.7 | 92.1 | 88.8 |
| VL2V-SD | 91.0 | 83.8 | 96.7 | 94.2 | 91.4 |
| Attn. Tune | 86.0 | 75.1 | 89.5 | 88.7 | 84.8 |
| Model Parameter Averaging (MPA) | 83.6 | 77.1 | 91.1 | 88.5 | 85.1 |
| CLIPOOD (Shu et al., 2023) | 92.3 | 81.3 | 96.0 | 94.2 | 90.9 |
| MIRO + SWAD | 90.4 | 83.2 | 95.5 | 92.8 | 90.5 |
| MIRO + MPA | 90.6 | 83.9 | 95.4 | 92.9 | 90.7 |

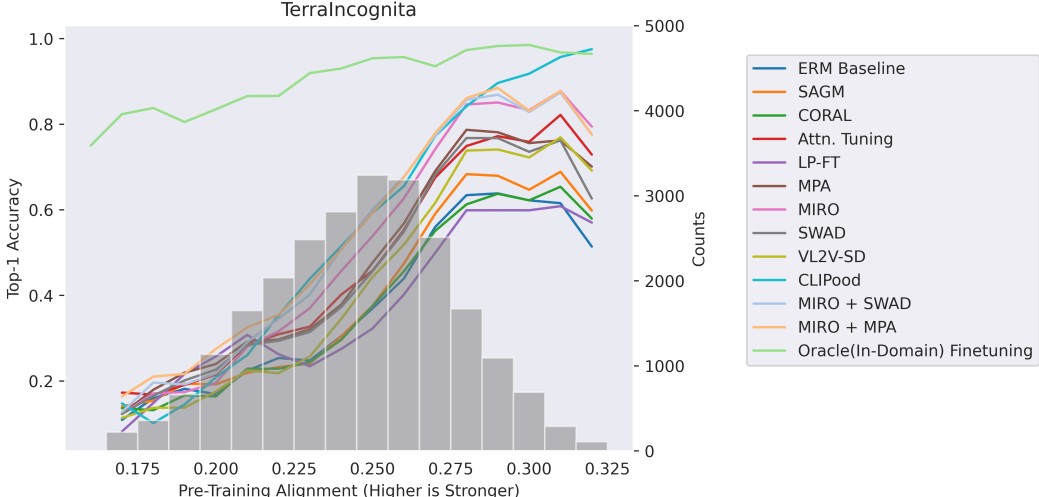

Figure 15: Plotting AlignmentScore vs Top-1 Accuracy on the TerraIncognita dataset.

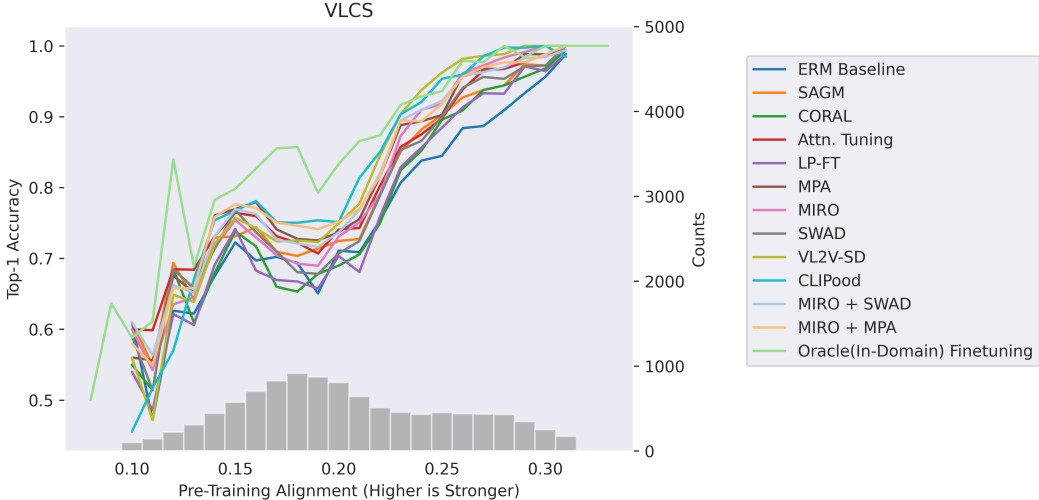

Figure 16: Plotting AlignmentScore vs Top-1 Accuracy on the VLCS dataset.

Table 12: Per-domain breakdown for OfficeHome (DomainBed-All)

| Method | Art | Clipart | Product | Real | Avg |
|---|---|---|---|---|---|
| OpenClip ZS | 83.4 | 73.2 | 92.4 | 92.6 | 85.4 |
| CORAL | 66.6 | 60.1 | 83.0 | 83.0 | 73.2 |
| SAGM | 70.7 | 67.1 | 84.3 | 83.6 | 76.4 |
| ERM* | 69.2 | 63.9 | 83.9 | 83.1 | 75.0 |
| LP-FT | 70.3 | 63.6 | 86.0 | 82.0 | 75.5 |
| SWAD | 76.6 | 74.3 | 89.0 | 87.2 | 81.8 |
| MIRO | 83.2 | 77.5 | 91.2 | 91.4 | 85.8 |
| VL2V-SD | 86.4 | 79.7 | 94.1 | 93.1 | 88.3 |
| Attn. Tune | 80.8 | 71.7 | 87.2 | 87.8 | 81.9 |
| Model Parameter Averaging (MPA) | 78.7 | 73.4 | 88.4 | 87.6 | 82.0 |
| CLIPOOD | 87.9 | 77.8 | 94.1 | 93.6 | 88.3 |
| MIRO + SWAD | 85.6 | 79.9 | 93.0 | 92.1 | 87.6 |
| MIRO + MPA | 85.6 | 80.5 | 93.0 | 92.3 | 87.9 |

Table 13: Per-domain breakdown for TerraIncognita (DomainBed-OOP)

| Method | L100 | L38 | L43 | L46 | Avg |
|---|---|---|---|---|---|
| OpenClip ZS | 5.1 | 0.0 | 6.4 | 6.6 | 4.5 |
| CORAL | 13.9 | 8.1 | 21.1 | 21.1 | 16.0 |
| SAGM | 17.2 | 7.8 | 20.2 | 32.2 | 19.3 |
| ERM* | 13.1 | 9.6 | 22.9 | 20.5 | 16.5 |
| LP-FT | 16.7 | 5.1 | 25.6 | 44.9 | 23.1 |
| SWAD | 20.7 | 8.2 | 26.9 | 28.2 | 21.0 |
| MIRO | 19.4 | 6.6 | 30.0 | 17.9 | 18.5 |
| VL2V-SD | 9.3 | 5.6 | 20.2 | 28.4 | 15.9 |
| Attn. Tune | 20.5 | 10.7 | 24.8 | 25.3 | 20.3 |
| Model Parameter Averaging (MPA) | 21.7 | 8.6 | 30.9 | 27.6 | 22.2 |
| CLIPOOD | 37.9 | 11.3 | 21.6 | 8.9 | 19.9 |
| MIRO + SWAD | 22.5 | 5.9 | 30.3 | 25.9 | 21.1 |
| MIRO + MPA | 24.8 | 9.6 | 32.5 | 32.8 | 24.9 |

Table 14: Per-domain breakdown for TerraIncognita (DomainBed-IP)

| Method | L100 | L38 | L43 | L46 | Avg |
|---|---|---|---|---|---|
| OpenClip ZS | 52.3 | 23.2 | 36.5 | 35.1 | 36.8 |
| CORAL | 46.4 | 35.3 | 56.9 | 32.8 | 42.9 |
| SAGM | 55.1 | 38.7 | 52.5 | 29.5 | 44.0 |
| ERM* | 37.4 | 38.0 | 55.2 | 37.5 | 42.0 |
| LP-FT | 50.7 | 36.5 | 58.1 | 18.2 | 40.9 |
| SWAD | 60.1 | 41.7 | 65.7 | 43.5 | 52.7 |
| MIRO | 69.5 | 50.2 | 69.7 | 46.1 | 58.9 |
| VL2V-SD | 52.8 | 38.4 | 57.6 | 43.8 | 48.1 |
| Attn. Tune | 52.4 | 44.0 | 68.0 | 47.5 | 53.0 |
| Model Parameter Averaging (MPA) | 63.8 | 41.4 | 67.9 | 44.7 | 54.4 |
| CLIPOOD | 77.7 | 56.4 | 69.0 | 51.0 | 63.5 |
| MIRO + SWAD | 68.9 | 54.8 | 73.4 | 51.2 | 62.1 |
| MIRO + MPA | 69.4 | 55.9 | 73.6 | 51.4 | 62.6 |

Table 15: Per-domain breakdown for TerraIncognita (DomainBed-All)

| Method | L100 | L38 | L43 | L46 | Avg |
|---|---|---|---|---|---|
| OpenClip ZS | 48.2 | 20.9 | 31.0 | 32.6 | 33.2 |
| CORAL | 43.6 | 32.7 | 50.4 | 31.8 | 39.6 |
| SAGM | 51.9 | 35.8 | 46.6 | 29.7 | 41.0 |
| ERM* | 35.3 | 35.3 | 49.3 | 36.0 | 39.0 |
| LP-FT | 47.8 | 33.5 | 52.1 | 20.5 | 38.5 |
| SWAD | 56.7 | 38.5 | 58.6 | 42.2 | 49.0 |
| MIRO | 65.2 | 46.0 | 62.4 | 43.7 | 54.3 |
| VL2V-SD | 49.1 | 35.3 | 50.8 | 42.5 | 44.4 |
| Attn. Tune | 49.7 | 40.8 | 60.1 | 45.6 | 49.1 |
| Model Parameter Averaging (MPA) | 60.1 | 38.3 | 61.1 | 43.2 | 50.7 |
| CLIPOOD | 74.3 | 52.1 | 60.4 | 47.4 | 58.5 |
| MIRO + SWAD | 64.9 | 50.1 | 65.5 | 49.0 | 57.4 |
| MIRO + MPA | 65.6 | 51.5 | 66.0 | 49.8 | 58.2 |

Table 16: Comparison of Methods for DomainNet Dataset (DomainBed-OOP)

| Method | Clp | Inf | Pnt | Qkdr | Real | Skt | Avg |
|---|---|---|---|---|---|---|---|
| OpenClip ZS | 29.2 | 43.5 | 28.7 | 0.0 | 35.5 | 21.2 | 26.3 |
| CORAL | 34.9 | 19.2 | 26.9 | 5.7 | 24.6 | 22.8 | 22.3 |
| SAGM | 36.4 | 19.4 | 25.1 | 6.1 | 26.8 | 24.2 | 23.0 |
| ERM* | 36.2 | 19.0 | 23.4 | 5.4 | 26.6 | 23.4 | 22.3 |
| LP-FT | 34.7 | 20.9 | 25.2 | 6.1 | 25.9 | 23.2 | 22.7 |
| SWAD | 42.5 | 28.3 | 33.4 | 7.6 | 30.2 | 29.5 | 28.6 |
| MIRO | 42.5 | 37.0 | 27.9 | 3.5 | 33.5 | 25.8 | 28.4 |
| VL2V-SD | 43.0 | 43.2 | 34.8 | 3.0 | 37.1 | 29.5 | 31.8 |
| Attn. Tune | 40.4 | 29.1 | 29.2 | 4.3 | 31.3 | 26.6 | 26.8 |
| Model Parameter Averaging (MPA) | 42.3 | 30.9 | 34.9 | 7.4 | 31.6 | 30.7 | 29.6 |
| CLIPOOD | 39.2 | 50.2 | 39.5 | 2.1 | 40.9 | 31.3 | 33.9 |
| MIRO + SWAD | 46.1 | 40.4 | 34.0 | 4.4 | 35.2 | 31.7 | 32.0 |
| MIRO + MPA | 46.2 | 42.6 | 36.0 | 4.4 | 36.0 | 33.3 | 33.1 |

Table 17: Comparison of Methods for DomainNet Dataset (DomainBed-IP)

| Method | Clp | Inf | Pnt | Qkdr | Real | Skt | Avg |
|---|---|---|---|---|---|---|---|
| OpenClip ZS | 84.0 | 84.1 | 85.5 | 23.5 | 91.4 | 81.1 | 74.9 |
| CORAL | 80.3 | 52.3 | 71.1 | 33.2 | 71.7 | 71.2 | 63.3 |
| SAGM | 81.7 | 52.8 | 70.2 | 34.5 | 73.4 | 73.1 | 64.3 |
| ERM* | 80.6 | 53.4 | 69.0 | 31.3 | 73.0 | 71.6 | 63.1 |
| LP-FT | 80.0 | 56.5 | 70.5 | 34.0 | 73.9 | 71.5 | 64.4 |
| SWAD | 85.5 | 67.1 | 80.2 | 41.3 | 80.1 | 79.8 | 72.3 |
| MIRO | 85.4 | 77.7 | 79.1 | 34.6 | 83.2 | 74.2 | 72.4 |
| VL2V-SD | 88.3 | 85.6 | 84.6 | 38.0 | 88.3 | 84.0 | 78.1 |
| Attn. Tune | 83.5 | 66.1 | 77.3 | 32.6 | 80.3 | 75.5 | 69.2 |
| Model Parameter Averaging (MPA) | 85.8 | 69.7 | 81.6 | 41.4 | 81.4 | 81.4 | 73.6 |
| CLIPOOD | 86.6 | 85.4 | 87.6 | 38.0 | 91.0 | 84.7 | 78.9 |
| MIRO + SWAD | 88.2 | 81.7 | 83.9 | 40.8 | 85.8 | 81.9 | 77.0 |
| MIRO + MPA | 88.1 | 83.7 | 85.5 | 41.3 | 86.5 | 84.0 | 78.2 |

Table 18: Comparison of Methods for DomainNet Dataset (DomainBed-All)

| Method | Clp | Inf | Pnt | Qkdr | Real | Skt | Avg |
|---|---|---|---|---|---|---|---|
| OpenClip ZS | 74.9 | 49.6 | 68.7 | 12.7 | 85.7 | 66.0 | 59.5 |
| CORAL | 72.7 | 27.1 | 58.0 | 20.1 | 67.1 | 58.8 | 50.6 |
| SAGM | 74.1 | 27.4 | 56.9 | 20.9 | 68.9 | 60.6 | 51.5 |
| ERM* | 73.1 | 27.4 | 55.7 | 19.0 | 68.5 | 59.3 | 50.5 |
| LP-FT | 72.4 | 29.3 | 57.1 | 20.7 | 69.2 | 59.2 | 51.3 |
| SWAD | 78.0 | 36.0 | 66.1 | 25.2 | 75.2 | 66.7 | 57.9 |
| MIRO | 78.0 | 42.9 | 64.0 | 20.0 | 78.3 | 61.7 | 57.5 |
| VL2V-SD | 80.5 | 47.8 | 69.5 | 21.6 | 83.1 | 69.8 | 62.0 |
| Attn. Tune | 76.1 | 36.0 | 63.0 | 19.2 | 75.4 | 62.9 | 55.4 |
| Model Parameter Averaging (MPA) | 78.3 | 37.9 | 67.4 | 25.2 | 76.5 | 68.2 | 58.9 |
| CLIPOOD | 78.4 | 52.9 | 72.6 | 21.2 | 85.9 | 70.7 | 63.6 |
| MIRO + SWAD | 80.8 | 45.6 | 68.8 | 23.7 | 80.7 | 68.7 | 61.4 |
| MIRO + MPA | 80.7 | 47.2 | 70.4 | 23.9 | 81.4 | 70.6 | 62.4 |

Table 19: Per-domain breakdown for VLCS (DomainBed-OOP)

| Method | Caltech101 | LabelMe | SUN09 | VOC2007 | Avg |
|---|---|---|---|---|---|
| OpenClip ZS | 100.0 | 62.0 | 77.1 | 81.6 | 80.2 |
| CORAL | 93.3 | 54.6 | 72.9 | 75.3 | 74.0 |
| SAGM | 80.0 | 58.8 | 76.5 | 78.0 | 73.3 |
| ERM* | 100.0 | 57.1 | 73.3 | 75.3 | 76.4 |
| LP-FT | 80.0 | 55.4 | 71.8 | 75.7 | 70.7 |
| SWAD | 100.0 | 51.2 | 79.1 | 77.7 | 77.0 |
| MIRO | 86.7 | 52.6 | 82.9 | 72.7 | 73.7 |
| VL2V-SD | 100.0 | 55.0 | 80.4 | 81.1 | 79.1 |
| Attn. Tune | 86.7 | 61.0 | 77.9 | 79.0 | 76.1 |
| Model Parameter Averaging (MPA) | 100.0 | 53.6 | 83.6 | 80.9 | 79.5 |
| CLIPOOD | 100.0 | 55.0 | 84.9 | 82.8 | 80.7 |
| MIRO + SWAD | 100.0 | 56.7 | 84.7 | 79.9 | 80.3 |
| MIRO + MPA | 100.0 | 54.6 | 83.4 | 77.4 | 78.9 |

Table 20: Per-domain breakdown for VLCS (DomainBed-IP)

| Method | Caltech101 | LabelMe | SUN09 | VOC2007 | Avg |
|---|---|---|---|---|---|
| OpenClip ZS | 99.8 | 91.0 | 94.8 | 97.8 | 95.9 |
| CORAL | 98.2 | 83.5 | 84.5 | 79.8 | 86.5 |
| SAGM | 95.9 | 84.3 | 86.3 | 85.5 | 88.0 |
| ERM* | 96.9 | 83.7 | 86.1 | 74.6 | 85.3 |
| LP-FT | 97.1 | 83.2 | 82.8 | 80.7 | 86.0 |
| SWAD | 97.9 | 82.9 | 88.4 | 84.8 | 88.5 |
| MIRO | 98.9 | 83.3 | 93.5 | 88.2 | 91.0 |
| VL2V-SD | 99.5 | 83.9 | 91.8 | 94.6 | 92.4 |
| Attn. Tune | 98.0 | 84.7 | 85.7 | 86.5 | 88.7 |
| Model Parameter Averaging (MPA) | 98.4 | 83.0 | 92.6 | 88.8 | 90.7 |
| CLIPOOD | 98.9 | 83.6 | 93.5 | 94.0 | 92.5 |
| MIRO + SWAD | 97.6 | 83.4 | 92.6 | 90.6 | 91.1 |
| MIRO + MPA | 98.0 | 83.2 | 92.9 | 90.1 | 91.0 |

Table 21: Per-domain breakdown for VLCS (DomainBed-All)

| Method | Caltech101 | LabelMe | SUN09 | VOC2007 | Avg |
|---|---|---|---|---|---|
| OpenClip ZS | 99.8 | 72.7 | 71.0 | 86.0 | 82.4 |
| CORAL | 98.2 | 66.2 | 72.7 | 77.0 | 78.5 |
| SAGM | 95.8 | 68.8 | 76.2 | 80.8 | 80.4 |
| ERM* | 96.9 | 67.8 | 73.0 | 73.9 | 77.9 |
| LP-FT | 97.0 | 66.4 | 71.4 | 77.4 | 78.0 |
| SWAD | 98.0 | 64.5 | 77.7 | 80.3 | 80.1 |
| MIRO | 98.7 | 65.3 | 81.2 | 79.3 | 81.1 |
| VL2V-SD | 99.5 | 66.6 | 78.8 | 86.0 | 82.7 |
| Attn. Tune | 97.9 | 70.3 | 77.2 | 81.9 | 81.8 |
| Model Parameter Averaging (MPA) | 98.4 | 65.7 | 81.3 | 83.7 | 82.3 |
| CLIPOOD | 98.9 | 66.7 | 81.6 | 86.6 | 83.4 |
| MIRO + SWAD | 97.6 | 66.4 | 81.6 | 82.6 | 82.0 |
| MIRO + MPA | 98.0 | 67.4 | 82.2 | 83.6 | 82.8 |

Table 22: Per-domain breakdown for PACS (DomainBed-OOP)

| Method | Art | Cartoon | Photo | Sketch | Avg |
|---|---|---|---|---|---|
| OpenClip ZS | 96.4 | 95.9 | 95.7 | 37.7 | 81.4 |
| CORAL | 80.4 | 84.3 | 89.5 | 42.2 | 74.1 |
| SAGM | 74.4 | 78.1 | 89.5 | 54.7 | 74.2 |
| ERM* | 75.9 | 86.8 | 91.6 | 53.2 | 76.9 |
| LP-FT | 74.2 | 85.9 | 97.9 | 56.3 | 78.6 |
| SWAD | 84.6 | 81.6 | 95.8 | 54.3 | 79.1 |
| MIRO | 94.2 | 94.1 | 97.9 | 52.8 | 84.7 |
| VL2V-SD | 93.5 | 96.5 | 97.9 | 52.2 | 85.0 |
| Attn. Tune | 94.0 | 91.4 | 95.8 | 55.7 | 84.2 |
| Model Parameter Averaging (MPA) | 94.0 | 88.9 | 97.9 | 50.2 | 82.7 |
| CLIPOOD | 96.0 | 96.8 | 97.9 | 58.3 | 87.2 |
| MIRO + SWAD | 96.0 | 94.9 | 100.0 | 50.7 | 85.4 |
| MIRO + MPA | 95.3 | 95.4 | 100.0 | 60.3 | 87.8 |

Table 23: Per-domain breakdown for PACS (DomainBed-IP)

| Method | Art | Cartoon | Photo | Sketch | Avg |
|---|---|---|---|---|---|
| OpenClip ZS | 99.7 | 99.7 | 99.9 | 94.6 | 98.5 |
| CORAL | 83.3 | 89.8 | 91.8 | 72.1 | 84.3 |
| SAGM | 89.7 | 93.0 | 94.7 | 82.9 | 90.1 |
| ERM* | 88.7 | 93.0 | 91.5 | 75.2 | 87.1 |
| LP-FT | 85.3 | 93.7 | 97.9 | 84.2 | 90.3 |
| SWAD | 96.4 | 93.0 | 97.8 | 91.0 | 94.6 |
| MIRO | 99.2 | 96.7 | 99.9 | 94.4 | 97.6 |
| VL2V-SD | 99.6 | 98.9 | 99.9 | 93.6 | 98.0 |
| Attn. Tune | 97.3 | 97.6 | 98.9 | 91.7 | 96.4 |
| Model Parameter Averaging (MPA) | 98.6 | 95.8 | 98.2 | 89.2 | 95.4 |
| CLIPOOD | 99.4 | 99.3 | 99.9 | 92.0 | 97.7 |
| MIRO + SWAD | 99.1 | 97.9 | 99.9 | 93.3 | 97.6 |
| MIRO + MPA | 99.0 | 98.5 | 100.0 | 94.7 | 98.1 |

Table 24: Per-domain breakdown for PACS (DomainBed-All)

| Method | Art | Cartoon | Photo | Sketch | Avg |
|---|---|---|---|---|---|
| OpenClip ZS | 97.8 | 98.7 | 99.8 | 91.6 | 97.0 |
| CORAL | 81.4 | 89.2 | 91.7 | 70.5 | 83.2 |
| SAGM | 83.4 | 90.4 | 94.6 | 81.4 | 87.5 |
| ERM* | 83.0 | 92.1 | 91.5 | 74.0 | 85.2 |
| LP-FT | 80.5 | 92.7 | 97.9 | 82.7 | 88.4 |
| SWAD | 91.5 | 91.2 | 97.8 | 89.0 | 92.4 |
| MIRO | 97.1 | 96.4 | 99.8 | 92.2 | 96.4 |
| VL2V-SD | 97.7 | 98.5 | 99.9 | 91.3 | 96.9 |
| Attn. Tune | 96.4 | 96.5 | 98.8 | 89.7 | 95.4 |
| Model Parameter Averaging (MPA) | 97.2 | 94.8 | 98.2 | 87.1 | 94.3 |
| CLIPOOD | 98.2 | 98.8 | 99.8 | 90.2 | 96.8 |
| MIRO + SWAD | 98.0 | 97.4 | 99.9 | 91.0 | 96.6 |
| MIRO + MPA | 97.8 | 98.0 | 99.9 | 92.9 | 97.2 |

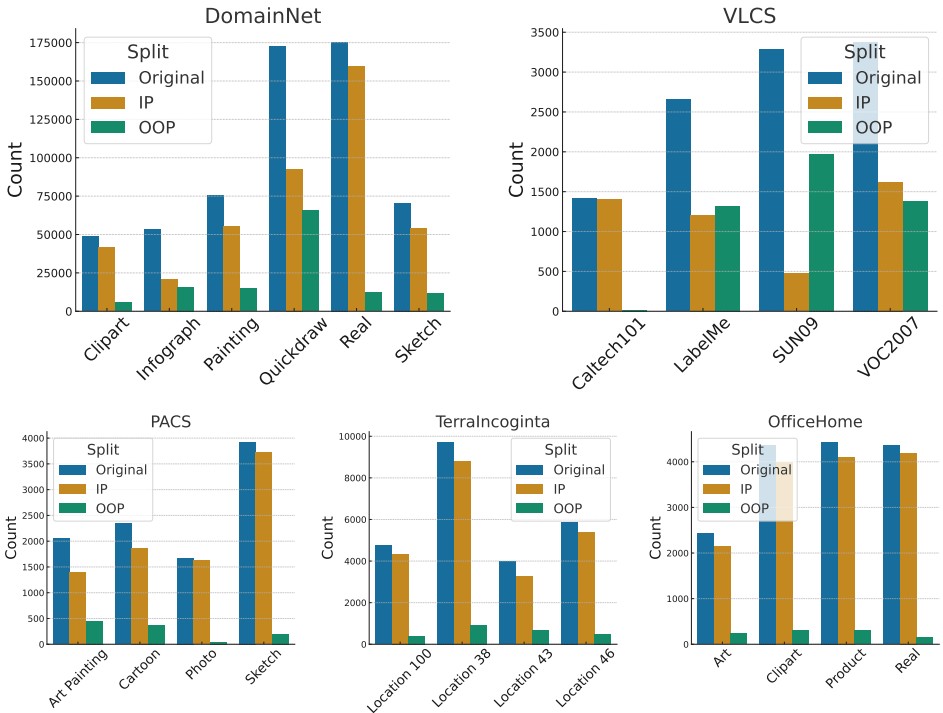

Figure 17: **DomainBed-(IP/OOP) Statistics**: Breakdown of DomainBed-IP and DomainBed-OOP counts, by dataset and domain. Overall, DomainNet and VLCS have the largest fraction of samples falling into DomainBed-OOP.

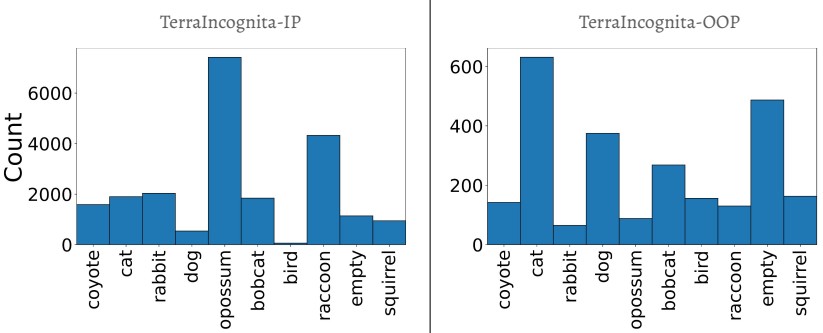

Figure 18: **Class-distribution shift:** TerraIncognita's class distribution differs between DB-IP and DB-OOP, indicating that some classes were better aligned during pretraining.

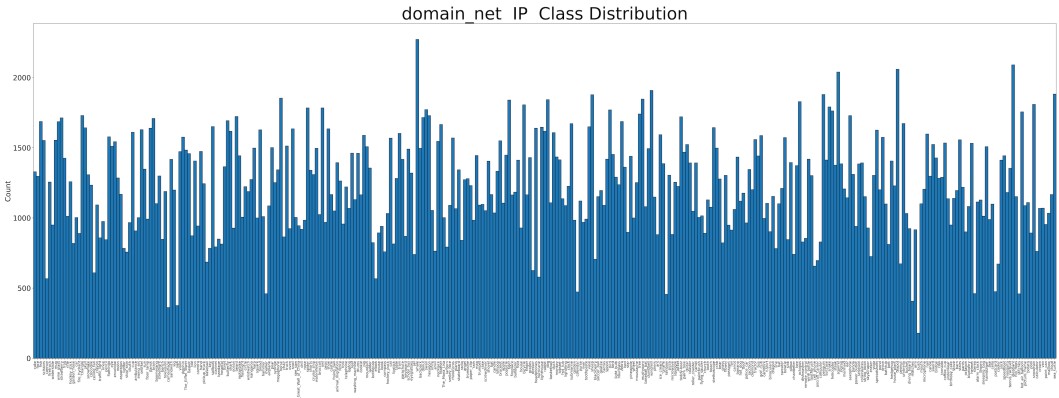

Figure 19: Class distribution of DomainNet-IP. Zoom in on pdf for best viewing.

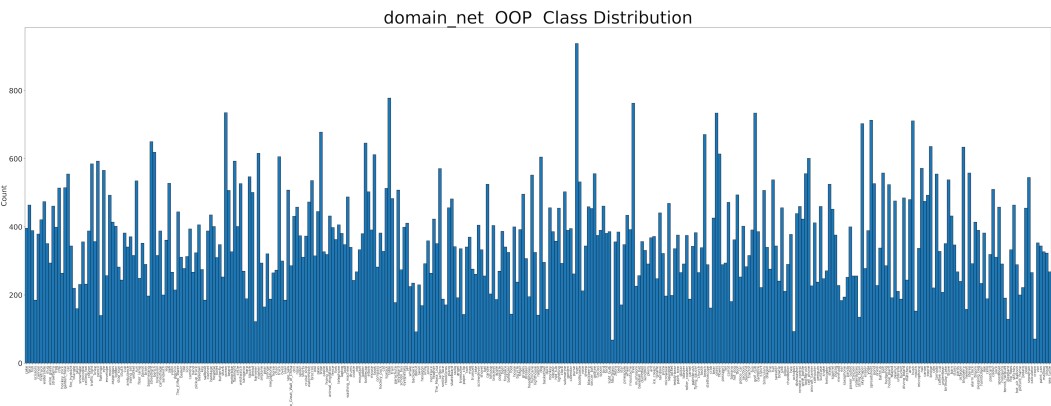

Figure 20: Class distribution of DomainNet-OOP. Zoom in on pdf for best viewing.

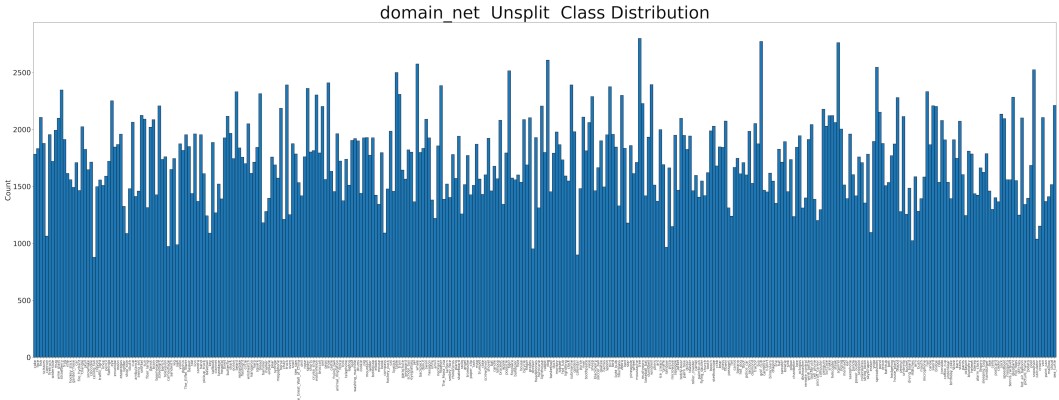

Figure 21: Class distribution of DomainNet before splitting. Zoom in on PDF before viewing.

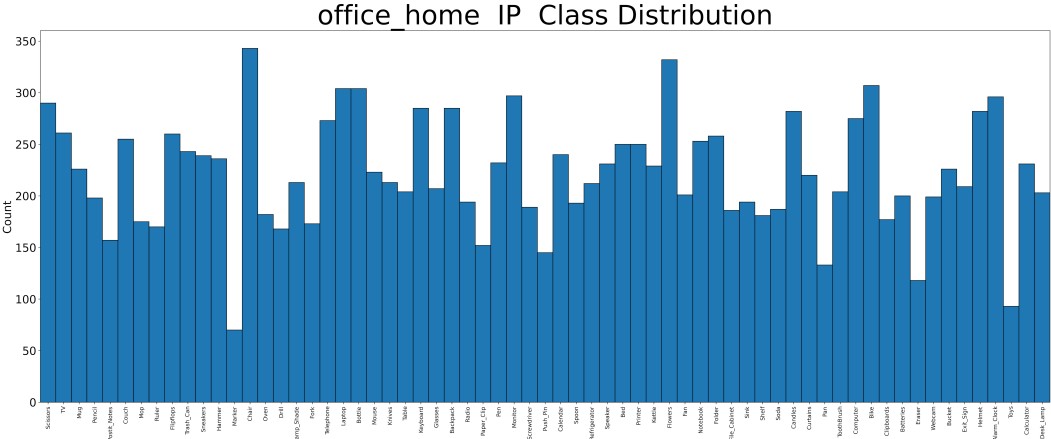

Figure 22: Class distribution of OfficeHome-IP. Zoom in on pdf for best viewing.

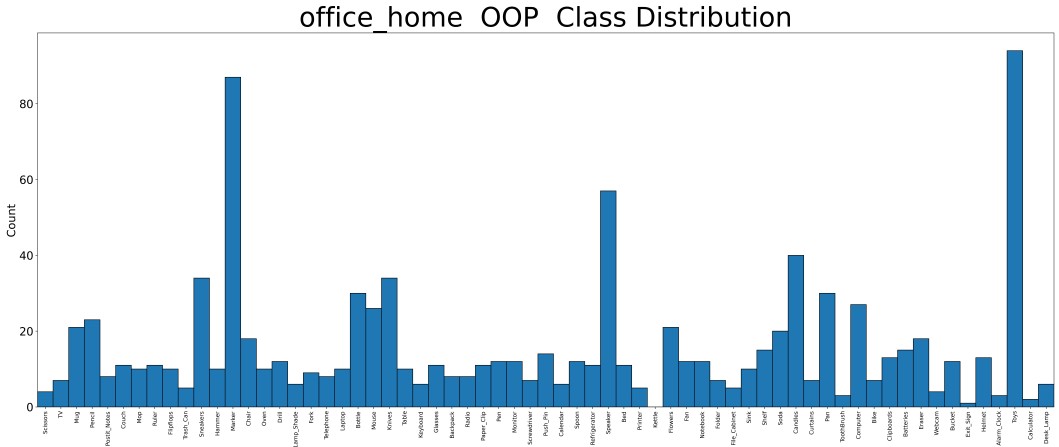

Figure 23: Class distribution of OfficeHome-OOP. Zoom in on pdf for best viewing.

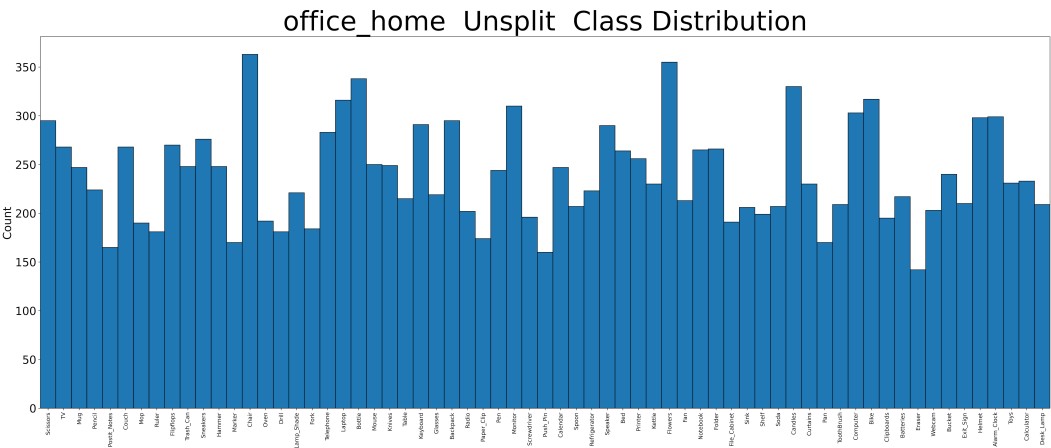

Figure 24: Class distribution of OfficeHome before Splitting. Zoom in on pdf for best viewing.

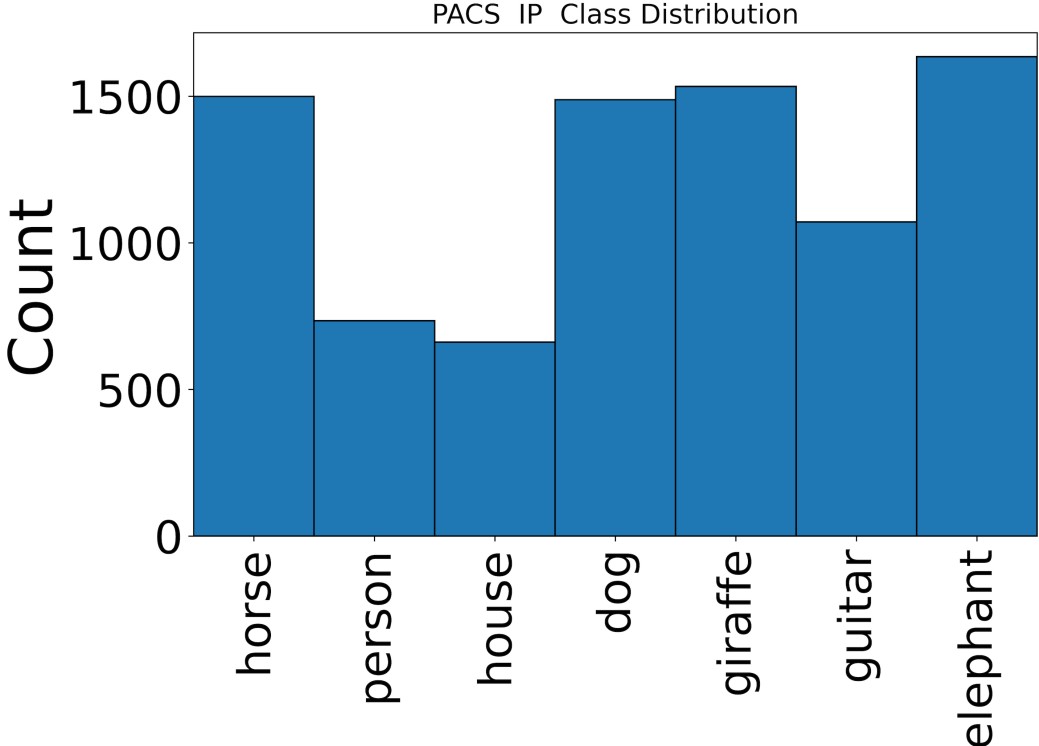

Figure 25: Class distribution of PACS-IP.

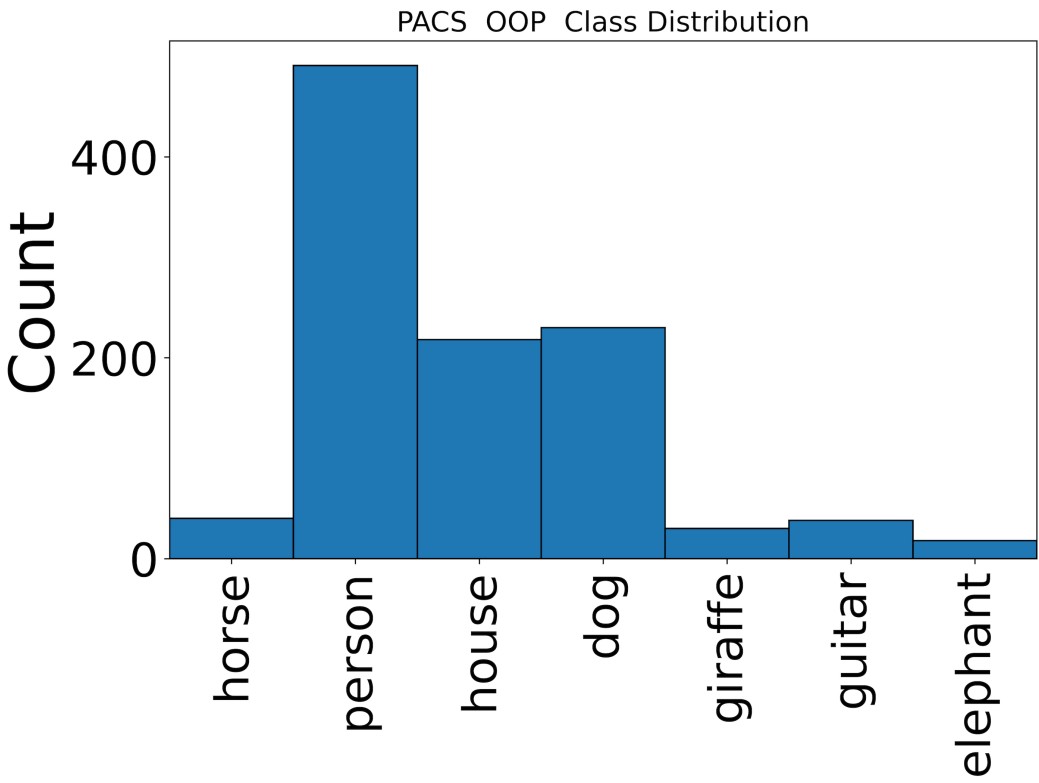

Figure 26: Class distribution of PACS-OOP.

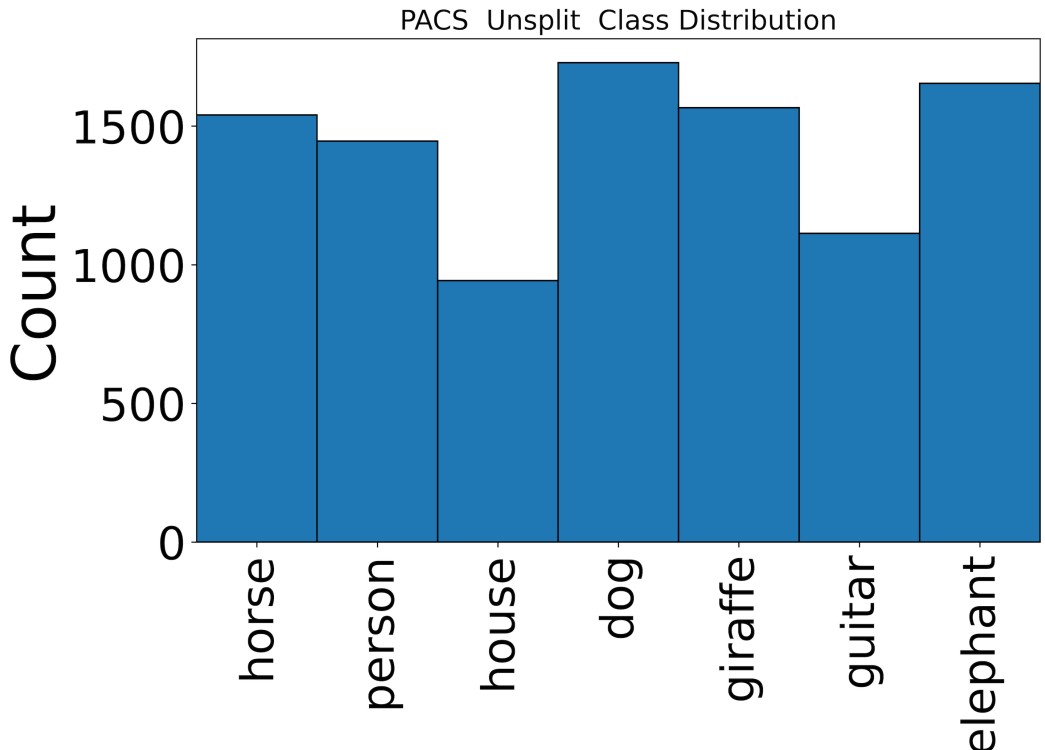

Figure 27: Class distribution of PACS before splitting.

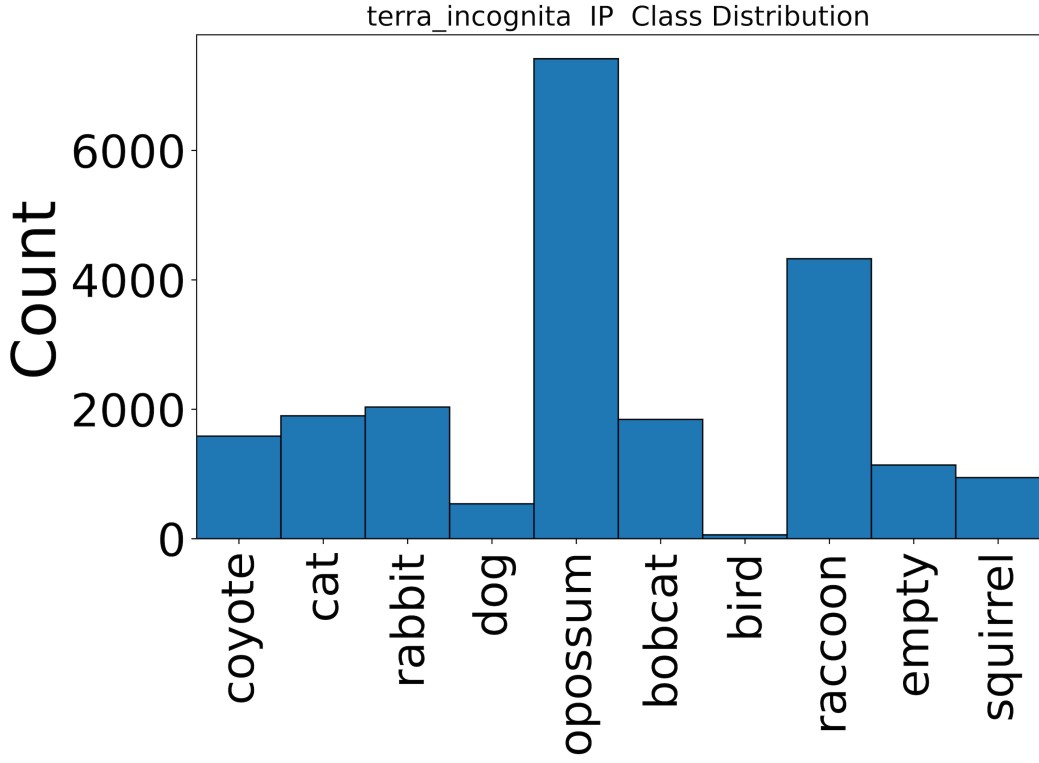

Figure 28: Class distribution of TerraIncognita-IP

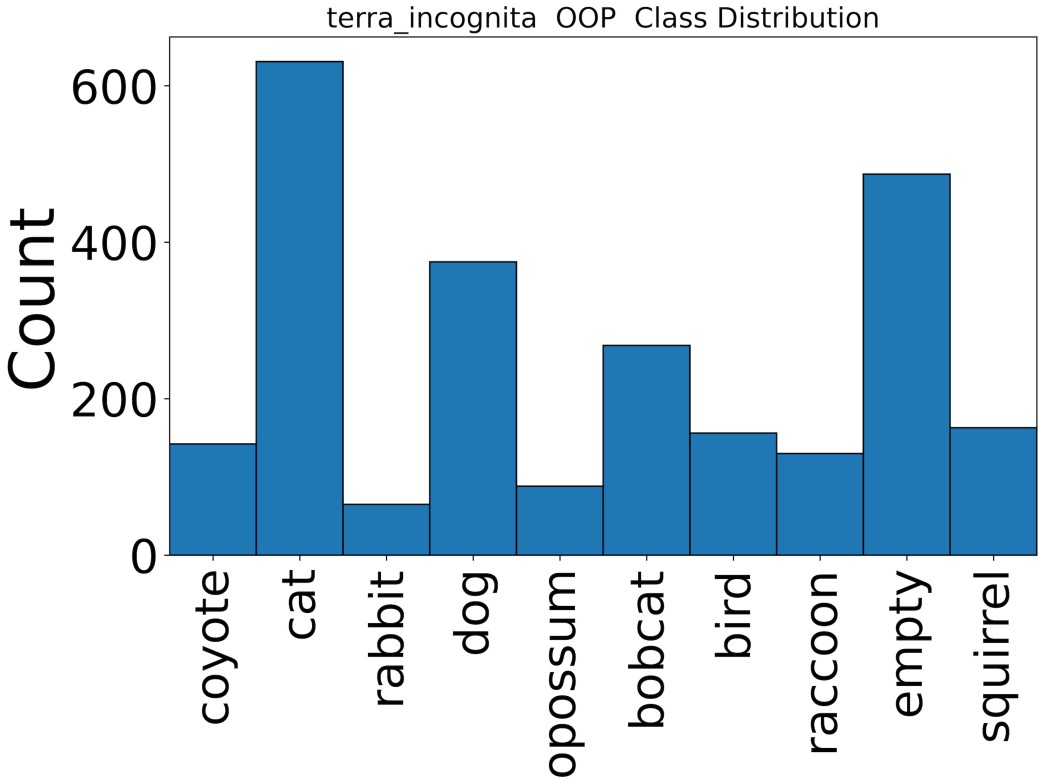

Figure 29: Class distribution of TerraIncognita-OOP

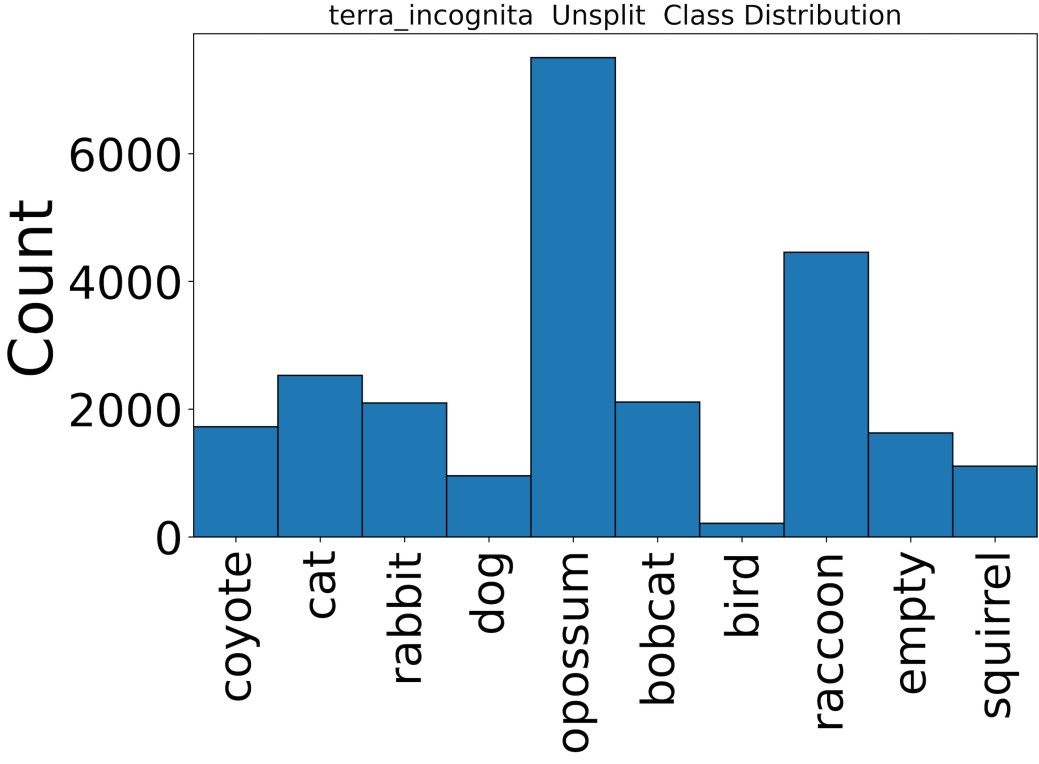

Figure 30: Class distribution of TerraIncognita before splitting.

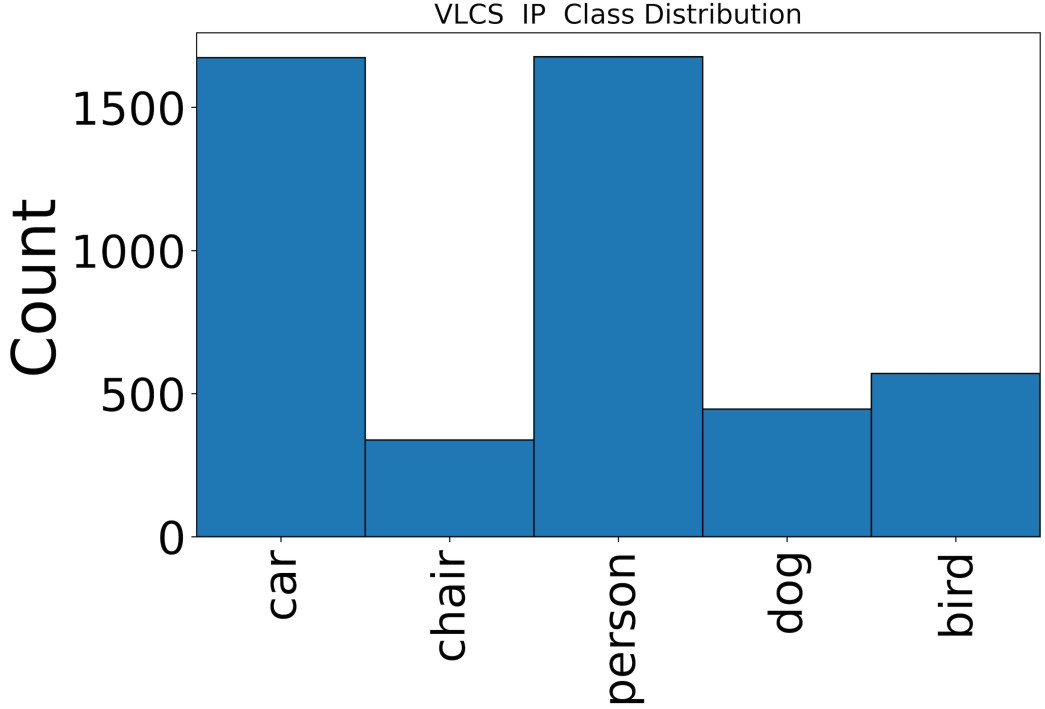

Figure 31: Class distribution of VLCS-IP

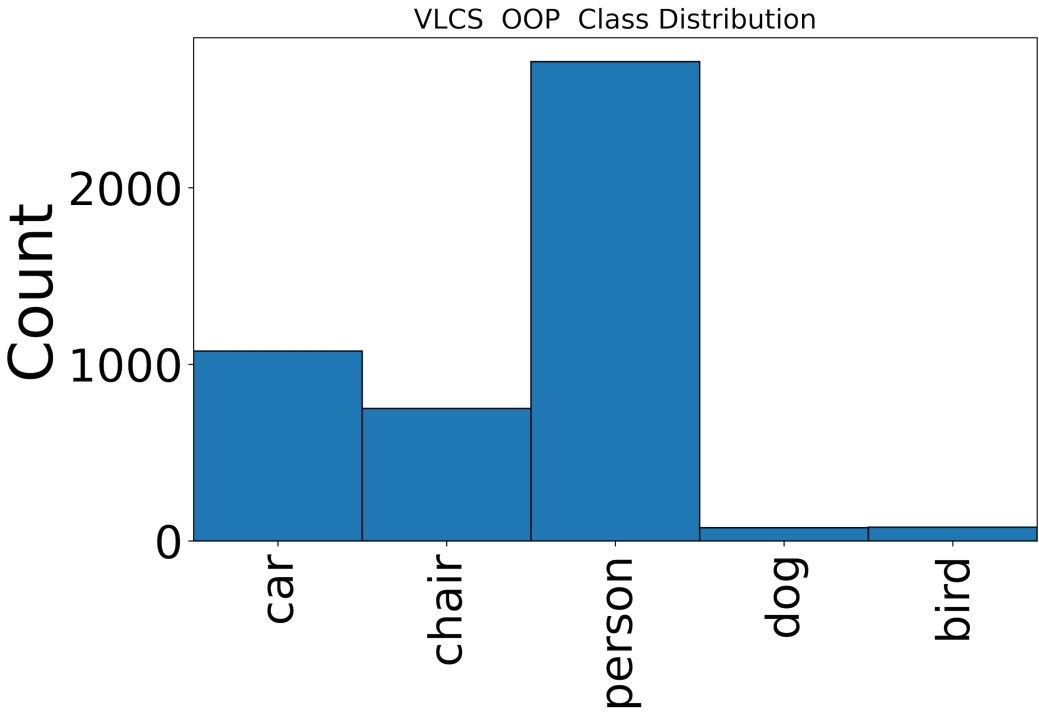

Figure 32: Class distribution of VLCS-OOP

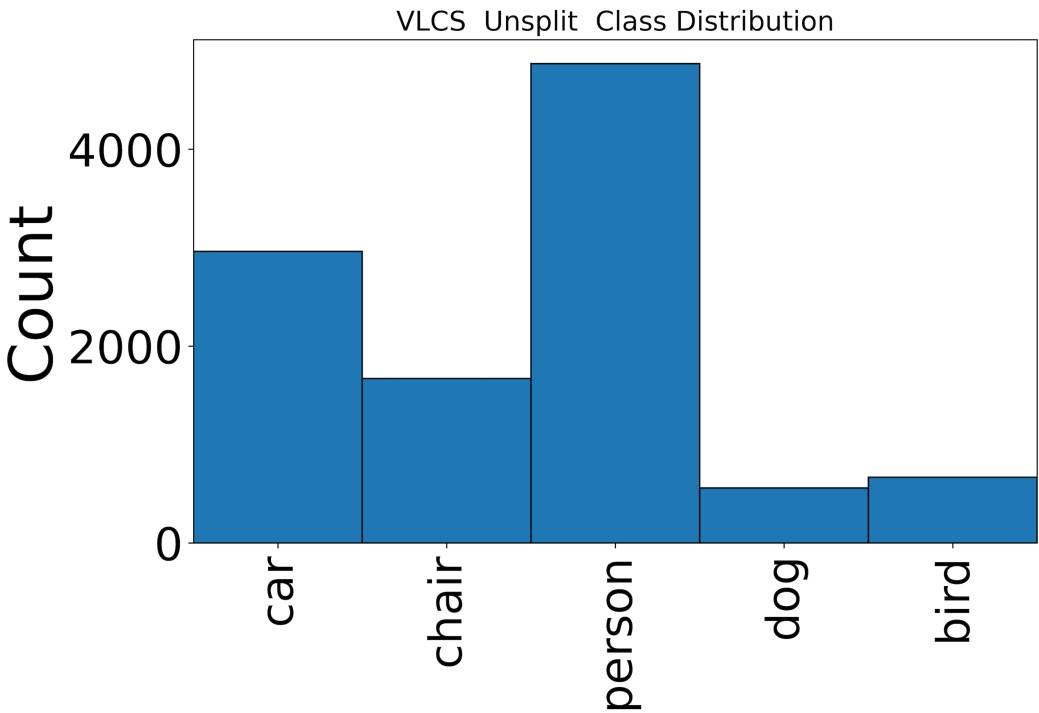

Figure 33: Class distribution of VLCS before splitting.

