# OpenReview forum: "Is Large-scale Pretraining the Secret to Good Domain Generalization?"
_ICLR.cc/2025/Conference — ICLR 2025 Poster_

### Official Review · Reviewer_DeUV · 2024-11-03

**Soundness:** 2
**Presentation:** 2
**Contribution:** 3
**Rating:** 6
**Confidence:** 4

**Summary:**

In this work, the authors investigate the relationship between certain aspects of pre-training and domain generalization (DG). They observe that high DG performance is achieved only when there is strong alignment between image and class label text embeddings. Based on this finding and evaluations on splits with lower alignment, they conclude that current DG methods predominantly rely on pre-trained features and struggle to learn novel, generalizable features from source domains.

**Strengths:**

1. The observation that alignment in pre-training between image and text embeddings significantly correlates to domain generalization (DG) performance is a valuable contribution.
2. The paper’s approach of dividing DomainBed datasets into In-Pretraining (IP) and Out-of-Pretraining (OOP) splits provides a novel lens for evaluating DG methods. This split reveals that while many models perform well on IP data, they struggle with OOP data, indicating that these models are not fully generalizing beyond pre-trained features.

**Weaknesses:**

Please see the questions section for more information.

**Questions:**

Although the central idea of the work is interesting, I have several concerns with the work as follows:
1. **CNN Evaluations**: It would be valuable to experiment with pre-trained CNNs and CNNs trained from scratch using some of the DG methods, then evaluate them on their corresponding IP/OOP splits. Do differences in IP/OOP performance persist when training CNNs from scratch as compared to adapting a pre-trained CNN? And therefore does a freshly trained CNN learn more generalizable features than a pre-trained one?

2. **Zero-Shot Classification**: To better assess how pre-training alignment influences DG performance, it would also be helpful to conduct zero-shot evaluations on both the IP and OOP test sets of DomainBed using the same backbone as the DG methods. This could open up various observations, such as the proportion of zero-shot classified points in each split and the number of points that shifted from correct to incorrect (or vice versa) after DG. What additional insights about alignment and generalization could zero-shot evaluation on IP and OOP data reveal?

3. **Revisiting the Image Similarity Hypothesis**: The image similarity hypothesis addresses the link between zero-shot generalization and similarity to the pre-training dataset but may not directly translate to DG. An analogous approach for DG might involve examining the perceptual similarity score between the target domain's evaluation set and the source domains' training set to see if it correlates with performance. Does this modified similarity hypothesis hold when applied to DG settings?

4. **Clarifying Key Takeaways**: While the paper presents some interesting insights, the main takeaways risk being obscured by details. A dedicated discussion section could enhance clarity by addressing questions such as (but not limited to): What insights from prior work connect pre-training with DG? What are the primary takeaways of this study, and how do they interrelate, and how would you contextualize with findings from prior work?

I am willing to adjust my score if these concerns are addressed adequately.

---

> ### Author Response · Authors · 2024-11-23
> **Rebuttal for Reviewer DeUV**
>
> We thank reviewer DeUV for their thoughtful comments, recognizing that the observation that alignment in pre-training between image and text embeddings significantly correlates to domain generalization (DG) performance is a valuable contribution.
>
> We answers the reviewers questions below:
>
> > 1. Training CNNs from scratch to evaluate if they learn more generalizable features:
>
> In our paper, we focus on analyzing the generalization capabilities of CLIP-based DG methods, as these have been demonstrated to be the strongest in recent work. However, we agree with the reviewer that from-scratch experiments are valuable, and so we also trained ResNet-50 models from scratch, using the MPA (Model Parameter Averaging) method and present the results below. We can see that performance is very low even on ALL DomainBed data (unsplit), further confirming that existing DG algorithms are  poor at learning new knowledge from source data alone, and they rely on strong pre-training for good performance.
> |                | DomainNet | OfficeHome | PACS | TI   | VLCS | Avg  |
> |----------------|-----------|------------|------|------|------|------|
> | Scratch        | 7.6       | 17.2       | 36.3 | 23.8 | 55.2 | 28.0 |
> | Clip ViT-B/16  | 58.9      | 82.0       | 94.3 | 50.7 | 82.3 | 73.6 |
>
>
> We add these results to Appendix B.9 (Table 5) in the updated pdf.
>
> > 2.  Zero-shot Classification Performance on DB IP/OOP:
>
> We thank the reviewer for suggesting a comparison to zero-shot performance, to better connect final DG performance to initial pre-training. We start by making a table of zero-shot performance over IP and OOP subsets, and comparing it to the strong DG method MIRO + MPA. We also compare to an upper bound, which is finetuned with ERM on the target distribution, and should therefore be considered an oracle.
>
> DB-IP
>
> |             | DomainNet | OfficeHome | PACS | TI   | VLCS | Avg  |
> |-------------|-----------|------------|------|------|------|------|
> | Zero-shot   | 74.9      | 89.0       | 98.5 | 36.8 | 95.9 | 79.0 |
> | MIRO+ MPA   | 78.2      | 90.7       | 98.1 | 62.6 | 91.0 | 84.1 |
> | Upper Bound | 81.6      | 88.5       | 97.8 | 93.4 | 93.8 | 91.0 |
> .
>
>
> DB-OOP
>
> |             | DomainNet | OfficeHome | PACS | TI   | VLCS | Avg  |
> |-------------|-----------|------------|------|------|------|------|
> | Zero-shot   | 26.8      | 48.1       | 81.4 | 4.5  | 80.1 | 48.2 |
> | MIRO + MPA  | 33.1      | 60.0       | 87.8 | 24.9 | 80.3 | 57.2 |
> | Upper Bound | 48.8      | 61.9       | 92.9 | 83.2 | 92.4 | 75.8 |
>
>
>
> DB-All
>
>
> |             | DomainNet | OfficeHome | PACS | TI   | VLCS | Avg  |
> |-------------|-----------|------------|------|------|------|------|
> | Zero-shot   | 59.6      | 85.4       | 97.0 | 33.2 | 82.4 | 71.5 |
> | MIRO + MPA  | 62.4      | 87.9       | 97.2 | 58.2 | 82.8 | 77.7 |
> | Upper Bound | 70.4      | 86.2       | 97.2 | 92.4 | 87.9 | 86.8 |
>
>
> We can make some interesting inferences. On the IP subset, the zero-shot model can achieve performances greater than the finetuned models (for PACS and VLCS).  This means that, for IP-data, DG finetuning sometimes causes more catastrophic forgetting than learning of new features, meaning MORE samples flip from correct to incorrect than the other way around! Overall, on average, there is a relatively small amount (~5%) to be gained from finetuning.
>
> In contrast, there is much to be gained from fine-tuning on the OOP subset. Performances are higher after fine-tuning for every dataset. This suggests that for data points not already aligned during pre-training, fine-tuning is critical. Yet, even the strongest evaluated method (MIRO + MPA) is very far from the upper bound! This suggests that finetuning is not effective enough at learning generalizable features beyond the pre-training alignment.
>
> We add zero-shot results to the main results  in Table 2.
>
> EDIT:
>
> We additionally compute confusion matrices for zero-shot and the  CLIPood DG method on IP and OOP subsets for the PACS dataset to better understand behaviour:
>
> **IP**
>
> | | DG: Incorrect | DG: Correct |
> |------------------|---------------|-------------|
>  | **ZS: Incorrect** | 141 | 70 |
> | **ZS: Correct**| 179 | 8231 |
>
>
> **OOP**
>
> | | DG: Incorrect | DG: Correct |
>  |------------------|---------------|-------------|
> | **ZS: Incorrect** | 98 | 62 |
> | **ZS: Correct** | 15 | 890 |
>
> One can see that DG methods flip very few correct samples to incorrect for OOP data, while they catastrophically forget more samples for IP data, suggesting that current DG methods  can even have negative value for IP data. However, even in the OOP case,  the SOTA CLIPood is not able to flip the majority of the incorrectly classified samples to correct for PACS. Overall, in both cases, CLIPood relies very heavily on pre-training. We add this analysis to Appendix B10.

---

> ### Author Response · Authors · 2024-11-23
> **Cont. Rebuttal for Reviewer DeUV**
>
> > 3.Recomputing the Image Similarity Hypothesis for source data instead of pre-training data:
>
> This is indeed an interesting question! As we state in the paper (Line 234 ), the main drawback of the Image Similarity hypothesis is that it does not consider how well the nearest perceptual neighbor is learned during pre-training. One reason for a sample being poorly learned during pre-training is that the pre-training caption is not very relevant to the DG task. Source data is unlikely  to have this issue, since source and target domains share labels.  Indeed when we compute perceptual similarity to source, presented in Appendix B.8(Figure 9) of the revised pdf,  we can see that there is a strong correlation between similarity and accuracy for the dataset OfficeHome, confirming the reviewers understanding.
>
>
>
>
> However, simply using source-data to compute the PerceptualSimilarity results in an incomplete understanding of the relationship between target data and the training procedure, due to the lack of consideration of the pre-training. For example, a sample which can gain very little from source finetuning (due to low similarity) but that is well aligned during pre-training can still achieve good DG performance. In fact, as we show in our response to question 2, not learning anything at all from source as in zero-shot evaluation, can result in quite strong performance for many datasets, underlining the importance of pre-training!
>
>
> > 4. Clarification of key takeaways:
>
>  We thank the reviewer for suggesting a dedicated discussion section in the paper.  We present it below, and it to the pdf on page 10:
>
> An increasing number of works in the Domain Generalization sub-field leverage pre-trained CLIP models for Domain Generalization benchmarks, it is important to better characterize the impacts of pre-training on DG. We leave the reader with the following takeaways:
>
> **Pre-training Alignment Predicts DG Performance**: Our study demonstrates that pre-training alignment, measured as the cosine similarity between image and text embeddings, is a robust predictor of DG performance. This holds true even after source fine-tuning, highlighting that the quality of alignment achieved during pre-training has a significant impact on the generalization capability of models.
>
> **Current DG Methods Exploit Pre-training Rather Than Learning New Features**: Our findings reveal a large difference in the performance of DG methods between pretraining-aligned (IP) and pretraining-misaligned (OOP) data. While state-of-the-art methods achieve near-oracle performance on IP data, they struggle significantly on OOP data. This indicates that current methods primarily leverage on pre-trained features rather than learning new, generalizable features from source data. Consequently, their success is heavily tied to the quality of pre-training, rather than the efficacy of the fine-tuning process itself.
>
> **Benchmarks Should Reflect Pre-training Reliance**: The reliance on pre-trained alignment calls for a reevaluation of DG benchmarks. Existing benchmarks often aggregate results across all target data, masking the limitations of DG methods on low-alignment samples. To address this, we propose splitting evaluation datasets into In-Pretraining (IP) and Out-of-Pretraining (OOP) subsets. This provides a clearer picture of where DG methods succeed and where they fail.  We hope that our proposed DomainBed-IP/OOP splits will guide the development of future methods that are better equipped to handle low-alignment data while maintaining performance on high-alignment samples.

---

> ### Author Response · Authors · 2024-11-26
> **Looking forward to a response**
>
> As the portion of the discussion period where updates to the pdf are allowed comes to a close, we would like to ask if there are any other modifications the reviewer would like us to make. We would be happy to answer any further questions as well.
>
> Thank you for the thoughtful review and looking forward to a response,
>
> Authors

---

> > ### Author Response · Authors · 2024-11-28
> > **Additional Zero-shot analysis**
> >
> > Dear reviewer,
> >
> > We have added additional confusion matrices to analyze zero-shot DG performance to question 2 in our rebuttal, and added the analysis to Appendix B.10.
> >
> > We welcome any additional questions from the reviewer!
> >
> > Authors

---

> > > ### Comment · Reviewer_DeUV · 2024-12-01
> > >
> > > Dear authors,
> > >
> > > I thank you for all the extra experiments and clarification. I now raise my score to acceptance.

---

> > > > ### Author Response · Authors · 2024-12-02
> > > > **Thank you**
> > > >
> > > > Thank you for taking the time to thoughtfully review our revisions and for your updated score. Your feedback is helpful in improving the paper.

---

### Official Review · Reviewer_R32V · 2024-11-03

**Soundness:** 3
**Presentation:** 2
**Contribution:** 3
**Rating:** 8
**Confidence:** 5

**Summary:**

This paper critically investigates the successes of recent DG methods that rely on pre-trained backbones, such as CLIP, with a focus on answering an important question: whether those successes are due to the effect of pre-training or to the algorithmic details of DG methods. To this end, the authors first consider the recently proposed image-similarity hypothesis, which posits that if the pre-training data contains images that are perceptually similar to the target data, then the zero-shot performance will be higher. However, the authors find that this does not fully explain the phenomenon and instead introduce the alignment hypothesis. They posit that if the alignment between image embeddings and text labels is high, with respect to the cosine similarity score defined in CLIP models, these alignment scores serve as a better indicator of generalization. They perform this analysis before fine-tuning and after fine-tuning, thus disentangling the effects of pre-training. Given this observation, they create a new benchmark that splits domain bed datasets into in-pretraining (IP) and out-of-pretraining (OOP) splits.

They then train a large number of DG algorithms from the domain bed framework and observe that most methods actually perform better in the IP condition and perform poorly on the OOP condition, where the performance gap between DG and oracle is quite high. They propose to release these datasets to provide the community with a new benchmark that encourages algorithmic innovations to improve generalization, rather than just using a more powerful backbone.

**Strengths:**

The paper is well-written and considers an important question of understanding the importance of pre-training backbones in modern DG approaches. Creating a new benchmark is an important contribution.

**Weaknesses:**

The paper could be improved, especially in Section 3. The methods for identifying image similarity scores and alignment scores need clearer explanations instead of the brief sentences currently provided. I suggest including an algorithmic table, as this is a key contribution.

While I appreciate the current contribution, it seems limited since it only considers one variant of the DG problem. The community has explored a broader range of generalization tasks  including sub-population shifts. It will be important to discuss how this work fits into those contexts as well. Additionally, the current alignment score relies solely on cosine similarity loss. I believe it should also take into account the uncertainties exhibited by the model, which would help in making a better split. This consideration is important because it would factor in the calibration of the pre-trained models, which the current approach does not address.

Finally, it is not very clear to me whether combining both image similarity and alignment scores would provide a better indicator or does alignment score already take that into account.
Also, I would recommend the authors toning down the statements like "DG has been solved with datasets like PACS etc". May be the takeaway that authors want to communicate is that some datasets should be dropped from future DG research and I think that is a better way to put it than the current style. As a community, DG needs new evaluation benchmarks that go beyond the now simple datasets and the excellent performance on these are hardly surprising given they have common categories that a pre-trained model would have been trained on.

Typos:

In the section 3.1, the equation of  perceptual similarity score is missing $\langle$.

Line 368, it should be ``between IP and OOP`` instead of  ``between OP and OOP``

**Questions:**

Please see weaknesses

---

> ### Author Response · Authors · 2024-11-23
> **Rebuttal for R32V**
>
> We thank the reviewer for their comments, and for pointing out the importance of the question we are studying, and the value of creating a new benchmark. We answer reviewer questions below.
>
> >  The paper could be improved, especially in Section 3. The methods for identifying image similarity scores and alignment scores need clearer explanations instead of the brief sentences currently provided. I suggest including an algorithmic table, as this is a key contribution.
>
> We thank the reviewer for this suggestion. We add an algorithm table to Section 3 as recommended.
>
> >  While I appreciate the current contribution, it seems limited since it only considers one variant of the DG problem. The community has explored a broader range of generalization tasks including sub-population shifts. It will be important to discuss how this work fits into those contexts as well.
>
> Thank you for bringing this up. Multi-source domain generalization is a particularly difficult form of subpopulation shift, where the testing population is not at all present in the training population. However, even with other forms of sub-population shift, there is reliance on foundation models for initialization[1] and an expectation to learn generalizable features from data with a different distribution than test data .Therefore, due to these similarities  we speculate that our findings would still hold in those scenarios.  We update the paper to briefly discuss subpopulation shift in line 112.
>
> [1]Dehdashtian, Sepehr, Lan Wang, and Vishnu Naresh Boddeti. "Fairerclip: Debiasing zero-shot predictions of clip in rkhss." International Conference on Learning Representations. 2024.
>
>
> >  Additionally, the current alignment score relies solely on cosine similarity loss. I believe it should also take into account the uncertainties exhibited by the model, which would help in making a better split. This consideration is important because it would factor in the calibration of the pre-trained models, which the current approach does not address.
>
>
> Thank you for the suggestion. In order to compute an alignment score which takes into account uncertainties, we compute a score using the confidence of the zero-shot classifier formed by the pre-trained CLIP model for each sample. Specifically, for a sample with  ground truth class c, we calculate the softmax over the logits output by the zero-shot classifier, and use the resulting probability p(c) as the score.  We refer to this as the Calibrated AlignmentScore and show the results in Appendix B.6 (Figure 6) of the updated manuscript.  We compare it to our Alignment Score, also in Appendix B.6 (Figure 7) of the updated manuscript.
>
> The reviewers suggestion smooths the curve, a desirable property. However, the scores have different scales across different datasets, making them very difficult to compare to each other in absolute terms. In contrast, the standard AlignmentScore does align across datasets to a greater degree. Either way, the story remains the same, DG methods cannot succeed for samples with low scores but are exceptionally strong for samples with high scores.
>
> >  Finally, it is not very clear to me whether combining both image similarity and alignment scores would provide a better indicator or does alignment score already take that into account.
>
> We compare 3 different scores on a subset of DomainNet data. The first, shown in blue in Appendix B.7 of the main paper (Figure  8) , is the AlignmentScore used in our work. The second, shown in orange, is the PerceptualSimilarity score. The last is a combination of the two, computed as an average, shown in green. As one can see this combination does not add any easily discernible signal to the AlignmentScore.

---

> ### Author Response · Authors · 2024-11-23
> **Cont. Rebuttal for R32V**
>
> >   Clarification of Takeaways
>
> We have added a new section about clarification of takeaways on Page 10, and have toned down statements about DG being solved for PACS in section 5.3. We present the new takeaways section below:
>
>
> “As an increasing number of works in the Domain Generalization sub-field leverage pre-trained CLIP models for Domain Generalization benchmarks, it is important to better characterize the impacts of pre-training on DG. We leave the reader with the following takeaways:
>
> **Pre-training Alignment Predicts DG Performance**: Our study demonstrates that pre-training alignment, measured as the cosine similarity between image and text embeddings, is a robust predictor of DG performance. This holds true even after source fine-tuning, highlighting that the quality of alignment achieved during pre-training has a significant impact on the generalization capability of models.
>
> **Current DG Methods Exploit Pre-training Rather Than Learning New Features**: Our findings reveal a large difference in the performance of DG methods between pretraining-aligned (IP) and pretraining-misaligned (OOP) data. While state-of-the-art methods achieve near-oracle performance on IP data, they struggle significantly on OOP data. This indicates that current methods primarily leverage on pre-trained features rather than learning new, generalizable features from source data. Consequently, their success is heavily tied to the quality of pre-training, rather than the efficacy of the fine-tuning process itself.
>
> **Benchmarks Should Reflect Pre-training Reliance**: The reliance on pre-trained alignment calls for a reevaluation of DG benchmarks. Existing benchmarks often aggregate results across all target data, masking the limitations of DG methods on low-alignment samples. To address this, we propose splitting evaluation datasets into In-Pretraining (IP) and Out-of-Pretraining (OOP) subsets. This provides a clearer picture of where DG methods succeed and where they fail.  We hope that our proposed DomainBed-IP/OOP splits will guide the development of future methods that are better equipped to handle low-alignment data while maintaining performance on high-alignment samples.”
>
>
> >  Typos
>
> Thank you, we have fixed them in the draft.

---

> > ### Author Response · Authors · 2024-11-26
> > **Looking forward to a response**
> >
> > We would like to update the reviewer that we have further slighltly updated the discussion of other types of domain generalization and  in the related works (line 114).
> >
> > As the portion of the discussion period where updates to the pdf are allowed comes to a close, we would like to ask if there are any other modifications the reviewer would like us to make. We would be happy to answer any further questions as well.
> >
> > Thank you for the thoughtful review and looking forward to a response,
> >
> > Authors

---

> > > ### Comment · Reviewer_R32V · 2024-12-01
> > > **THank you for the response**
> > >
> > > I am satisfied by the responses and additional experiments and improvements to the paper for enhanced clarity. I am hence recommending acceptance.

---

> > > > ### Author Response · Authors · 2024-12-02
> > > > **Thank you**
> > > >
> > > > Thank you for carefully reviewing our revisions and for your updated score. We truly appreciate your feedback in helping us improve the paper.

---

### Official Review · Reviewer_BJqk · 2024-11-04

**Soundness:** 2
**Presentation:** 2
**Contribution:** 2
**Rating:** 5
**Confidence:** 5

**Summary:**

The paper proposes a hypothesis stating that if the test images and their corresponding text labels (e.g., "A photo of a {cls}") are already well-aligned in the embedding space before fine-tuning, the model's domain generalization (DG) performance will be high after fine-tuning.

Using an alignment score, the target dataset is divided into two parts: in-distribution (IP) and out-of-distribution (OOP). The experiments show that DG methods perform well on high-alignment samples (DomainBed-IP) after fine-tuning but poorly on low-alignment samples (DomainBed-OOP). This suggests that current DG methods rely heavily on pre-trained features.

**Strengths:**

I appreciate that the authors consider the impact of pre-training weights in the current DG evaluation protocol.

I also value their effort to remove noisy data labels in standard DG benchmarks.

This paper presents a comprehensive set of experiments.

**Weaknesses:**

**Weaknesses**

- Since the goal of CLIP's contrastive loss is to maximize the similarity between the ground truth text label embedding and image embedding, it's unsurprising that the alignment score (measured as similarity to the ground truth text embedding) correlates with final performance after fine-tuning. Additionally, the alignment score applies only to vision-language models, not to pure vision models, which is a limitation that should be mentioned in the paper.

- When comparing the predictive power of the Alignment and Image Similarity Hypotheses for downstream tasks, it would be helpful to provide quantitative metrics showing how strongly each hypothesis correlates with performance.

- In line 177, is there a typo where “before” should be “after” in “The cosine similarity between image and ground truth text-label embeddings before pre-training”? Also, just to confirm, are the alignment scores and similarity scores shown in Figure 2 calculated from the CLIP model immediately after pre-training on LAION, but before fine-tuning on the downstream DG benchmark?

- I wonder if using a Perceptual Similarity Score with a well-chosen threshold could similarly divide the data into high-similarity samples (DomainBed-IP) and low-similarity samples (DomainBed-OOP). Would the same experimental results hold—that DG methods perform well on DomainBed-IP but struggle on DomainBed-OOP after fine-tuning?

- Based on experiments showing that DG methods perform well on high-alignment samples (DomainBed-IP) but struggle on low-alignment data (DomainBed-OOP), the paper concludes that current DG methods rely on pre-trained features. While this is a useful insight, the conclusion is somewhat unsurprising, as it’s well-known that downstream task performance heavily depends on the diversity of the pre-trained dataset, whether in-distribution or out-of-distribution. For those interested in accurately measuring a DG method’s generalization performance, [1] suggests a new evaluation protocol isolating pre-training's impact. Alternatively, pre-training on a larger and more diverse dataset is likely the best approach for those focused on improving downstream generalization.

[1] Rethinking the Evaluation Protocol of Domain Generalization. CVPR2024

**Questions:**

I have several questions regarding the significance of the proposed Perceptual Similarity Score and the conclusions of this paper. Please see the details above.

---

> ### Author Response · Authors · 2024-11-23
> **Rebuttal for reviewer  BJqk**
>
> We thank reviewer BJqk for their helpful comments. We thank them for their recognition of our effort to remove noisy data labels in standard DG benchmarks, and our set of comprehensive experiments.
>
> We answer the reviewers questions below:
>
> >  Is it surprising that DG methods do not improve beyond pre-training?
>
> If Domain Generalization finetuning methods were very effective, alignment before finetuning would be less strongly correlated with DG performance after finetuning. This is because training on source domains would learn generalizable features from source data where pre-trained features are insufficient. This is what DG methods are supposed to do, so it is surprising that they do not. Of course, Domain Generalization is a fundamental problem which is very difficult, so some relationship between pre-training and final performance is expected. However, our stance is that better DG methods in the future may be able to combine strong pre-training with effective learning from source. We believe our work is the first building block for such a direction.
>
> >  Alignment score only applies to Vision-Language Models, a limitation which should be mentioned:
>
> Thank you for bringing this up. We mention it in line 138 of the submission, and updated the draft to discuss it in the introduction in line 45. We do believe that extending to non-vision language models is very interesting future work.
>
> > When comparing the predictive power of the Alignment and Image Similarity Hypotheses for downstream tasks, it would be helpful to provide quantitative metrics showing how strongly each hypothesis correlates with performance.
>
> Thank you for the suggestion. We compute the Pearson correlation for the Perceptual Similarity Score, which tests the Image Similarity Hypothesis, and find it to be 0.84. For the Alignment Score, which tests our Alignment Hypothesis, the correlation is 0.91. This indicates that the Alignment Hypothesis has a stronger correlation with performance. However, our goal goes beyond just correlation—we show that samples with very low AlignmentScore cannot generalize well at all while those with low Perceptual Similarity Score can still generalize to some degree (see Figure 2 b.) in the main paper).
>
> >  In line 177, is there a typo where “before” should be “after” in “The cosine similarity between image and ground truth text-label embeddings before pre-training”?
>
> Yes, thanks for catching this! We updated the draft with this change.
>
> >  Also, just to confirm, are the alignment scores and similarity scores shown in Figure 2 calculated from the CLIP model immediately after pre-training on LAION, but before fine-tuning on the downstream DG benchmark?
>
> Yes, this is the correct understanding.
>
> >  I wonder if using a Perceptual Similarity Score with a well-chosen threshold could similarly divide the data into high-similarity samples (DomainBed-IP) and low-similarity samples (DomainBed-OOP). Would the same experimental results hold—that DG methods perform well on DomainBed-IP but struggle on DomainBed-OOP after fine-tuning?
>
> Actually, based on Figure 2b.) one can see that the slope of the line showing Top-1 Accuracy vs Perceptual Similarity Score is positive, but has a relatively shallow slope. This suggests that the performance difference across samples with varying similarity scores is not very large. Then, splitting the data using perceptual similarity will not yield very different results in the two splits.
>
> In contrast, Figure 2a.) show that the slope of Top-1 Accuracy vs AlignmentScore is very steep, which gives us the ability to split into DomainBed-IP and DomainBed-OOP effectively.
>
> As an example, we split at a PerceptualSimilarityScore value of 0.86. We can see the differences are not very large between OOP and IP, indicating that AlignmentScore is a better thresholding metric.
>
>
> |     | PACS | VLCS | TerraIncognita | OfficeHome | DomainNet |
> |-----|------|------|----------------|------------|-----------|
> | IP  | 97.2 | 74.8 | 60.6           | 86.8       | 61.4      |
> | OOP | 95.6 | 77.4 | 42.9           | 78.3       | 55.3      |
>
> We add this result to Appendix B.5 (Table 4).

---

> > ### Author Response · Authors · 2024-11-23
> > **Cont. Rebuttal for Reviewer BJqk**
> >
> > >  It’s well-known that downstream task performance heavily depends on the diversity of the pre-trained dataset, whether in-distribution or out-of-distribution. For those interested in accurately measuring a DG method’s generalization performance, [1] suggests a new evaluation protocol isolating pre-training's impact. Alternatively, pre-training on a larger and more diverse dataset is likely the best approach for those focused on improving downstream generalization.
> >
> > Thank you for sharing [1], this is very interesting work. We added it to our discussion of related works in line 152.  While [1] suggests NOT initializing with large-scale vision-language foundation models, we agree with the reviewer that leveraging such models remains critical for improved generalization performance, so excluding pre-training from the evaluation is not always desirable. However, as we say in our concluding arguments (line 537), foundation models will always have distributional tails where they fail.  Being able to leverage additional source data, even when it does not perfectly align with the target data, is important in this scenario.  We believe our DomainBed-IP/OOP splits enable research where the goal is to preserve robust features where they exist, and learn them better from source where they do not.

---

> > > ### Author Response · Authors · 2024-11-26
> > > **Looking forward to a response**
> > >
> > > As the portion of the discussion period where updates to the pdf are allowed comes to a close, we would like to ask if there are any other modifications the reviewer would like us to make. We would be happy to answer any further questions as well.
> > >
> > > Thank you for the thoughtful review and looking forward to a response,
> > >
> > > Authors

---

### Official Review · Reviewer_Zx21 · 2024-11-05

**Soundness:** 3
**Presentation:** 3
**Contribution:** 3
**Rating:** 6
**Confidence:** 3

**Summary:**

This paper analyzes the reliance of DG works on pre-training particularly in the context of CLIP.  The authors propose the Alignment hypothesis which states that pre-trained alignment between image and class embeddings is helpful for domain generalization performance prediction even after a model is finetuned on source domains. For evaluation, the authors utilize a two-step method. First, they cleaned the data. Images with AlignmentScore < 0.15 are discarded. Similarly, high AlignmentScore (>0.4) is usually for samples that have label text in the image. Second, In-pretraining and Out-of-pretraining are determined based on a scoring threshold of 0.21, based on the authors' observations about the data.

**Strengths:**

The paper is well-written and generally easy to follow.

This paper presents some interesting insights:

  1. Relation of low AlignmentScore and presence of label noise in the domain generalization datasets.

  2. "We find that all methods, including those considered state-of-the-art, perform poorly on OOP data" which seems to suggest that DG methods do not learn anything beyond what has already been learned by CLIP during pre-training. This also aligns with some of the previous findings.

  3. Model parameter averaging boosting performance.

  4. The fact that a high score indicates text in the image, which is not a significant part of the data (0.00-0.15% across the dataset).

**Weaknesses:**

IP/OOP method of splitting data seems to be a bit circular. Some of the results are a bit difficult to interpret. For instance, Table 2 seems to represent the correlation of final performance and prediction with IP/OOP splits but presentations make it hard to understand. Further concerns are mentioned in the questions.

**Questions:**

- IP/OOP splitting by using CLIP AlignmentScore does not make sense. The CLIP model is first used to calculate to split the data and then to predict the score. Is not it circular?

- One assumption behind this work is that simple alignment between image and label embeddings is sufficient to measure the presence of data in pre-training. What is the justification for this assumption? If a model has learned well, it should have high similarity for examples that are not part of training data but are in the training distribution.


- The authors state the following in the abstract: "Not just the presence of data in pre-training but how well is it learned", however, this part is not reflected later in the paper. How does IP/OOP reflect this?

---

> ### Author Response · Authors · 2024-11-23
> **Rebuttal For Reviewer Zx21**
>
> We thank the reviewer for their thoughtful comments, and appreciate that they believe the paper presents interesting insights. We answer questions below.
>
> > The CLIP model is first used to calculate to split the data and then to predict the score. Is not it circular?
>
> The CLIP model is used to split the data before DG finetuning, and the performance is evaluated after DG finetuning. This design avoids circularity because the splitting process only leverages the pre-trained embeddings to establish a baseline measure of alignment, while the evaluation assesses DG performance AFTER finetuning. DG methods start from pre-trained models but continue learning on source data, to learn additional generalizable features. If DG methods were highly effective at learning from source data, the performance would be high for both IP and OOP splits.
>
>
> > One assumption behind this work is that simple alignment between image and label embeddings is sufficient to measure the presence of data in pre-training. What is the justification for this assumption? If a model has learned well, it should have high similarity for examples that are not part of training data but are in the training distribution.
>
> Actually, we do NOT assume anything about the presence of training data in our work, and we agree with the reviewer on this point.  We instead make the claim that alignment is sufficient for predicting generalization, but don’t make assumptions of how or why alignment arose. Alignment is a function of all inputs to the pre-training process, including the training data, model architecture, and pre-training objectives. Although alignment can be high for images present in the pre-training, it need not be. For example, data points in the tail of the training distribution may not be well aligned at all.  On the flip side, as the reviewer points out, data in the training distribution but not part of the training data may be well aligned. Anecdotally, however, we do find examples of test - train contamination (Figure 4 in the Appendix) and speculate this may be a contributor to high alignment and performance.  We have updated the main paper to clarify this in line 53.
>
> > The authors state the following in the abstract: "Not just the presence of data in pre-training but how well is it learned", however, this part is not reflected later in the paper. How does IP/OOP reflect this?
>
> We assume the reviewer is referring to a statement in line 48 of the introduction (not abstract).  Alignment is our measure of how well a sample is learned. This means that a sample’s image embedding is aligned with the textual description of the content of the image, as defined by the ground truth label in the DG dataset.  Then, OOP splits are data which are not well learned, while IP splits are data which are well learned.  We verify that DG performance is higher for IP splits than OOP splits. We have updated the discussion in the introduction to clarify this, in line 82.
>
>
> > Presentation of results should be improved
>
> We have added a  Discussion section on page 10 to clarify key takeaways, and have added various discussions throughout the text. All new text is highlighted in red. We will continue to finetune and improve the presentation for the camera ready.

---

> ### Author Response · Authors · 2024-11-26
> **Looking forward to a response**
>
> We would like to update the reviewer that we have further updated the discussion of evaluations on IP/OOP to connect them to AlignmentScore, and therefore how well target samples are leared during pretraining in lines 464 and 299. We also more explicitly connected the Alignment Hypothesis to how well a sample is learned in line 52.
>
> As the portion of the discussion period where updates to the pdf are allowed comes to a close, we would like to ask if there are any other modifications the reviewer would like us to make. We would be happy to answer any further questions as well.
>
> Thank you for the thoughtful review and looking forward to a response,
>
> Authors

---

### Author Response · Authors · 2024-11-25
**Official Author Comment**

We thank the reviewers for their feedback.  We wanted to take a moment to summarize the reviews and our response, in addition to highlighting the fact that we are happy to answer any additional questions.
The reviewers identified several key strengths of our work:
- The introduction of the **Alignment Hypothesis** and its valuable insights into the role of pre-training alignment in DG performance (*Reviewer Zx21, Reviewer DeUV*).
 - The creation of the **IP/OOP benchmark**, which provides a methodology for evaluating DG methods and highlights their limitations (*Reviewer R32V, Reviewer DeUV*).
- Comprehensive experimental evaluations  of DG performance, and consideration of how pre-training weights impact evaluation(*Reviewer BJqk*).
-  Efforts to clean and refine DomainBed datasets, including addressing label noise (*Reviewer BJqk,Zx21*).

Responses to reviewer comments include:
- **Circularity (*Reviewer Zx21,BJqk*)**: We clarified the separation of splitting and evaluation phases to address concerns about circularity. The splitting process uses pre-trained embeddings for alignment, while evaluation assesses post-finetuning DG performance. Good DG methods would learn beyond their pre-training.

- **Additional evaluations(*Reviewer DEuV,BJqk,R32V*)**: We add a number of additional experiments. We show splitting DomainBed using PerceptualSimilarityScore is not as effective as AlignmentScore(*Reviewer BJqk*), compute correlations (*Reviewer BJqk*), compare with a score which includes uncertainty (*Reviewer R32V*), combine PerceptualSimilarity and Alignment Scores (*Reviewer R32V*), train CNNs from scratch (*Reviewer DeUV*), add zero-shot results (*Reviewer DeUV*), and do an analysis of the ImageSimilarityHypothesis with source (instead of pretraining) data (*Reviewer DeUV*). We present them in the per-reviewer comments, and also update the manuscript.

- **Takeaways and presentations(*Reviewers DeUV,  R32V,Zx21*)**: We have added a dedicated discussion section summarizing key takeaways from our findings and their implications for the DG community(*Reviewers DeUV,  R32V*). We have also made clarifications in the introduction regarding the assumptions of pre-training data and connecting the learning of pre-training to IP/OOP splits(*Reviewer Zx21*)


We  have updated the manuscript to include suggested changes (highlighted in blue).

---

### Author Response · Authors · 2024-11-26
**Light PDF Updates**

Dear reviewers,

We have continued revising the manuscript lightly. Changes are as follows:

* We have further slighltly updated the discussion of evaluations on IP/OOP to connect them to AlignmentScore, and therefore how well target samples are leared during pretraining in lines 464 and 299. We also more explicitly connected the Alignment Hypothesis to how well a sample is learned in line 52.

* We have further slighltly updated the discussion of other types of domain generalization and  in the related works (line 114).

We have also changed the revision color from red to blue to improve legibility.

Since the window for updating the pdf is coming to a close, we would like to ask the reviewers if they have any additional questions or requests in an updated manuscript.

---

### Meta-Review · Area_Chair_MbWZ · 2024-12-20

**Metareview:**

This paper investigates whether the successes of recent DG methods are due to the effect of pre-training or to the algorithmic details of DG methods. While it is an empirical study, it presents some interesting insights for this problem. Only one reviewer did not give a positive feedback, but the weakness was not very crucial -- an unsurprising conclusion is not necessarily a bad thing. Thus, I recommended an acceptance.

**Additional Comments On Reviewer Discussion:**

The reviewers agreed that authors' rebuttal addressed their concerns and raised the scores accordingly.

---

### Decision · Program_Chairs · 2025-01-22

Accept (Poster)